# TET1-mediated DNA hydroxymethylation regulates adult remyelination in mice

Sarah Moyon[1✉], Rebecca Frawley[1], Damien Marechal[1], Dennis Huang[1], Katy L. H. Marshall-Phelps[2], Linde Kegel[2], Sunniva M. K. Bøstrand [2], Boguslawa Sadowski[3,4,5], Yong-Hui Jiang[6], David A. Lyons [2], Wiebke Möbius [3,4,5] & Patrizia Casaccia [1,7✉]

The mechanisms regulating myelin repair in the adult central nervous system (CNS) are unclear. Here, we identify DNA hydroxymethylation, catalyzed by the Ten-Eleven-Translocation (TET) enzyme TET1, as necessary for myelin repair in young adults and defective in old mice. Constitutive and inducible oligodendrocyte lineage-specific ablation of *Tet1* (but not of *Tet2*), recapitulate this age-related decline in repair of demyelinated lesions. DNA hydroxymethylation and transcriptomic analyses identify TET1-target in adult oligodendrocytes, as genes regulating neuro-glial communication, including the solute carrier (*Slc*) gene family. Among them, we show that the expression levels of the $Na^+/K^+/Cl^-$ transporter, SLC12A2, are higher in *Tet1* overexpressing cells and lower in old or *Tet1* knockout. Both aged mice and Tet1 mutants also present inefficient myelin repair and axo-myelinic swellings. Zebrafish mutants for *slc12a2b* also display swellings of CNS myelinated axons. Our findings suggest that TET1 is required for adult myelin repair and regulation of the axon-myelin interface.

[1] Neuroscience Initiative Advanced Science Research Center, New York, NY, USA. [2] Centre for Discovery Brain Sciences, Edinburgh, UK. [3] Department of Neurogenetics, Göttingen, Germany. [4] Electron Microscopy Core Unit, Max-Planck-Institute of Experimental Medicine, Göttingen, Germany. [5] Center Nanoscale Microscopy and Molecular Physiology of the Brain (CNMPB), Göttingen, Germany. [6] Department of Neurobiology and Pediatrics, Duke University School of Medicine, Durham, NC, USA. [7] Program of Biology and Biochemistry, The Graduate Center of The City University of New York, New York, NY, USA. ✉email: smoyon@gc.cuny.edu; pcasaccia@gc.cuny.edu

New myelin formation in the adult brain is critical for repair of damaged or lost myelin, which is impaired in several neurological and psychiatric disorders[1–4]. Myelin is the specialized membrane of oligodendrocytes (OLs), whose differentiation from oligodendrocyte progenitor cells (OPCs) results from the interplay of transcription factors and epigenetic regulators that can be influenced by diverse external stimuli[5,6]. Most of these mechanisms have been studied in the context of developmental myelination or in primary cultured cells[7–10]. However, the epigenetic marks in each cell type, are affected by external conditions, age and disease states[3,11–15]. We and others have reported the role of several epigenetic marks at distinct stages of oligodendrocyte cell lineage and discussed the importance of specific chromatin modulators for developmental myelination[8,10,16,17] and remyelination[18,19]. In particular, DNA methylation, catalyzed by DNA methyltransferases (DNMTs), is the single most significant epigenetic mark predictive of biological age[14–20]. We recently reported an age-dependent role of DNMT1 regulating proliferation and differentiation of neonatal OPCs (nOPCs)[16] and having only minor effects on the differentiation of adult OPCs (aOPCs) during myelin repair[19]. However, our genome-wide DNA methylation dataset was obtained using bisulfite-conversion sequencing[16], and therefore could not distinguish between two distinct modifications on cytosine residues, with opposing transcriptional outcomes: DNA methylation (5mC), an epigenetic modification generally associated with transcriptional repression, and DNA hydroxymethylation (5hmC), mainly associated with transcriptional activation[21,22]. Together with the distinct roles of DNMTs in developmental myelination and adult remyelination, this raised the question on the genome-wide distribution and functional role of DNA hydroxymethylation in adult myelin repair, that is addressed in this study.

DNA hydroxymethylation refers to the oxidation of methylated cytosine residues in the DNA (5mC), in a multi-step reaction catalyzed by enzymes called Ten-Eleven Translocation (TETs)[21–25]. These enzymes act as methyl-cytosine dioxygenases, which convert a hydrogen atom at the C5-position of cytosine to a hydroxymethyl group (5hmC), through oxidation of 5mC[23–25]. As such, this reaction has been suggested to serve as an intermediate step to promote DNA demethylation and favor gene expression[21,22,24–27]. The TET family includes proteins that are particularly enriched in brain tissue and differentially expressed in neuronal and glial cells[21,26–28]. In neuronal progenitors, TETs are necessary for their differentiation during cortical development[29], and later for axonal growth and functional neuronal circuits formation[30,31]. In glial cells, although their expression has been reported in cultured glial cells[28] and dysregulated in glioblastomas[32], a thorough molecular and functional analysis of DNA hydroxymethylation in the adult brain and in myelin repair is lacking.

Here, we combined unbiased sequencing approaches and in silico data analysis, with immunohistochemical and phenotypic characterization of lineage specific constitutive and inducible mouse mutants to identify TET1 as an enzyme responsible for the expression of genes regulating the axon–myelin interface during myelin repair.

## Results

### Age-dependent DNA hydroxymethylation during myelin regeneration.
To start characterizing the distinct contribution of DNA methylation and hydroxymethylation to spinal cord OLIG2 + oligodendroglial cells at postnatal day 7 (P7) and adulthood (P60), we assessed nuclear immunoreactivity to antibodies specific for the DNA methylation mark 5 methylcytosine (5mC)

(Fig. 1a, b) or for the DNA hydroxymethylation mark 5 hydroxymethylcytosine (5hmC) (Fig. 1c, d). The levels of immunoreactivity were expressed as the ratio of intensity of either 5mC or 5hmC to the nuclear area of OLIG2 + cells (expressed in pixels/$\mu m^2$) and the proportion of OLIG2 + cells with low (<5 pixels/$\mu m^2$), medium (between 5 and 10 pixels/$\mu m^2$) and high (>10 pixels/$\mu m^2$) was calculated. The proportion of OLIG2 + cells with high levels of DNA methylation increased from $4.33 \pm 1.23$ in the neonatal spinal cord to $28.84 \pm 3.75\%$ in the adult tissue (Fig. 1b). The time-dependent increase in the proportion of OLIG2 + cells with high levels of DNA hydroxymethylation was even more striking, ranging from $1.84 \pm 1.03\%$ in the neonatal spinal cord to $35.50 \pm 8.12\%$ in the adult tissue (Fig. 1d). Since hydroxymethylation requires the presence of a methyl group on the DNA as substrate, the higher proportion of OLIG2 + cells with high levels of 5mC and 5hmC in the adult compared to the neonatal spinal cord, set the premise for a deeper investigation on the functional role of this mark in adult myelin formation.

As a major function of adult OPCs is myelin repair in response to a demyelinating injury, we opted to use the lysolecithin (LPC)-induced model of demyelination in the spinal cord of young (P60) and old (P540) adult mice (Fig. 1e). The selection of this experimental model was due to the fact that the temporal pattern of demyelination and remyelination has been extensively reported to decline with age[33–38]. Consistent with previous reports[37], remyelination after LPC injection was efficient in young P60 adult mice 21 days after lesion (21dpl), and impaired in old P540 mice, as indicated by the lower number of remyelinated axons (Fig. 1f, g). The age-dependent decline in remyelination was also assessed on cryosections, using Fluoromyelin staining at 7, 14, and 21dpl (Supplementary Fig. 1a–d). In young adults, the intensity of the Fluoromyelin staining gradually recovered after lesion, reaching the same levels of intensity as unlesioned white matter tracts by 21dpl (Supplementary Fig. 1a, b), while the recovery was delayed and less effective in older mice (Supplementary Fig. 1c, d). The successful recovery in young mice was preceded by a progressive increase in the percentage of OLIG2 + cells with high levels of 5hmC starting at 14dpl (Fig. 1h and Supplementary Fig. 1e). Old mice, in contrast, showed a delayed and impaired response, characterized by a lower percentage of OLIG2 + cells with high 5hmC (Fig. 1i), although the overall density of OLIG2 + cells remained constant between young and old mice (Supplementary Fig. 1f). As previously described, adult OPC in young and old mice started proliferating in response to a demyelinating lesion, although the response was less prominent in old mice (Supplementary Fig. 1g). Differentiation of OLIG2 + cells into CC1 + oligodendrocytes, was also much more pronounced in young mice (Fig. 1j), than old ones (Fig. 1k). Together, these data support an overall age-dependent decline of DNA hydroxymethylation and remyelination.

### TET1 and TET2 are the most abundant isoforms in the oligodendrocyte lineage, although only TET1 shows an age-dependent decline.
DNA hydroxymethylation is catalyzed by the Ten-Eleven Translocation (TET) family of enzymes, which includes TET1, TET2, and TET3. We therefore started characterizing the presence of these distinct isoforms by performing immunohistochemistry in cryosections of adult spinal cord (Fig. 2a) and corpus callosum (Fig. 2b). Of the three enzymes, TET1 and TET2 showed clear immunoreactivity in OLIG2 + cells in the adult spinal cord, with TET3 being detected only in a very small percentage of OLIG2 + cells (Fig. 2c). To further address the impact of aging on the levels of these enzymes, we sorted OPCs from neonatal (P5), young (P60) and old (P540) *Pdgfra-H2BEGFP* reporter mice, and used a single-cell suspension capture method, based on microfluidics (C1 Fluidigm), to amplify transcripts by

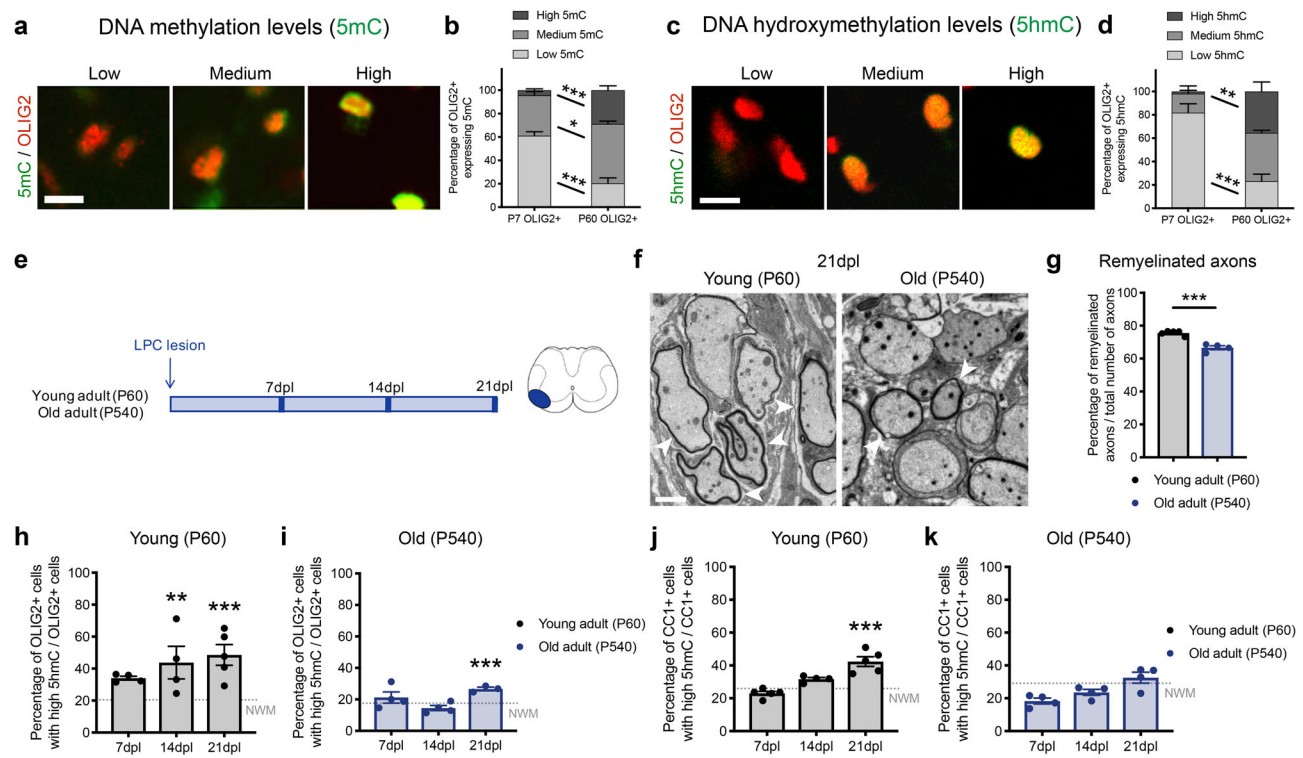

**Fig. 1 Age-dependent DNA hydroxymethylation during new myelin formation. a** Nuclear immunoreactivity specific for 5 methylated cytosine (5mC, green) and OLIG2 (red) in adult mouse spinal cord sections. Scale bar = 10 μm. **b** Bar graphs showing the relative proportion of OLIG2 + cells with low, medium, or high levels of 5mC immunoreactivity in the neonatal (P7) and adult (P60) spinal cord. Error bars represent SEM for n = 4 mice (*p < 0.05 and ***p < 0.001, two-way ANOVA, for age and 5mC immunoreactivity). **c** Nuclear immunoreactivity specific for hydroxymethylated cytosines (5hmC, green) and for OLIG2 (red) in adult mouse spinal cord sections. Scale bar = 10 μm. **d** Bar graphs showing the relative proportion of OLIG2 + cells with low, medium, or high levels of 5hmC immunoreactivity in the neonatal (P7) and adult (P60) spinal cord. Error bars represent SEM for n = 4 mice (**p < 0.01 and ***p < 0.001, two-way ANOVA, for age and 5hmC immunoreactivity). **e** Schematic experimental timeline of lysolecithin lesion experiment in young and old mice. **f** Electron micrographs of lesioned spinal cord sections from young (P60) and old (P540) mice at 21 days post-lesion (21dpl). Remyelination in young and old adult samples shown by white arrowheads. Scale bar = 5 μm. **g** Data indicate the average remyelinated axons in young and old spinal cord 21dpl. Error bars represent SEM for n = 5 young mice and n = 4 old mice (***p = 0.0002, Student's t test two-tailed). **h, i** Quantification of the percentage of OLIG2 + cells with high levels of 5hmC in spinal cord lesions at 7dpl, 14dpl, and 21dpl, in young (**h**) and old (**i**) mice. Dotted line indicates the relative percentage of OLIG2 + cells with high level of 5hmC in unlesioned white matter tracts (NWM). Data represent the average of 3 sections/mouse. Error bars represent SEM for n = 5 young mice and n = 4 old mice (**p < 0.01 and ***p < 0.001, one-way ANOVA, for time after lesion). **j, k** Quantification of the percentage of CC1 + cells with high level of 5hmC in spinal cord lesions in young (**j**) and old (**k**) mice analyzed at 7dpl, 14dpl, and 21dpl. Dotted line indicates the relative percentage of CC1 + cells with high level of 5hmC in unlesioned white matter tracts (NWM). Data points indicate the average 3 sections/mouse. Error bars represent SEM for n = 5 young mice and n = 4 old mice (***p < 0.001, one-way ANOVA, time after lesion).

real-time quantitative PCR (rt-qPCR) of the three preparations (Fig. 2d). Transcripts for *Tet1* (Fig. 2e) and the percentage of TET1 expressing OLIG2 + cells (Fig. 2f) were the most dramatically decreased by aging. *Tet2* transcript levels and the percentage of TET2 expressing OLIG2 + cells, in contrast, were only minimally impacted by age (Fig. 2e, f). The age-dependent differences in TET1 (Supplementary Fig. 2a–d), but not TET2 (Supplementary Fig. 2e–h) levels, persisted in the lesioned adult spinal cord during repair from demyelinating lysolecithin injections (Supplementary Fig. 2). Together, these findings identify the levels of TET1, and not of TET2, to decline with age, concurrently with the levels of 5hmC in oligodendroglial lineage cells.

**Constitutive or inducible ablation of *Tet1* in OPCs mimics the events leading to defective myelin regeneration detected in old mice.** As aging resulted in the progressive decline of *Tet1*, we asked whether a lineage specific constitutive conditional knockout for *Tet1* would mimic the inefficient remyelination phenotype observed in old mice. We therefore crossed *Olig1-cre* mice[39] with the *Tet1-flox* line (gift from Pr. Yong-Hui Jiang)[40] (Fig. 3) or with

the *Tet2-flox* line[41] to further address the specificity of the isoform (Supplementary Fig. 3). Both crosses generated truncated and catalytically inactive TET1 or TET2 isoforms (see Methods for details). Survival, body size, and weight of *Olig1^{cre/+};Tet1^{fl/fl}* (*Tet1* mutants) or *Olig1^{cre/+};Tet2^{fl/fl}* (*Tet2* mutants) mice did not differ from their respective controls (*Olig1^{+/+};Tet1^{fl/fl}* and *Olig1^{+/+};Tet2^{fl/fl}*) (Fig. 3a–c, Supplementary Fig. 3a–c) and no overt motor phenotype was detected.

Immunohistochemistry validated the cell lineage specificity of *Tet1* (Supplementary Fig. 4a) or *Tet2* (Supplementary Fig. 4b) ablation in OLIG2 + cells, and not in ISLET1 + neurons (Supplementary Fig. 4c, d) or GFAP + astrocytes (Supplementary Fig. 4e, f). We also validated the absence of any compensatory upregulation of other TET isoforms in each of the mutants, compared to controls (Fig. 3d, Supplementary Fig. 3d).

Quantification of immunostained spinal cord (Fig. 3e, h) from *Tet1* mutants did not reveal any difference in OLIG2 + or CC1 + cell density (Fig. 3f) or in the extent of MBP + areas (Fig. 3g), compared to controls. Quantification of myelin protein levels in P60 spinal cord extracts did not show significant differences between the two genotypes (Fig. 3i, j). Ultrastructural analysis of

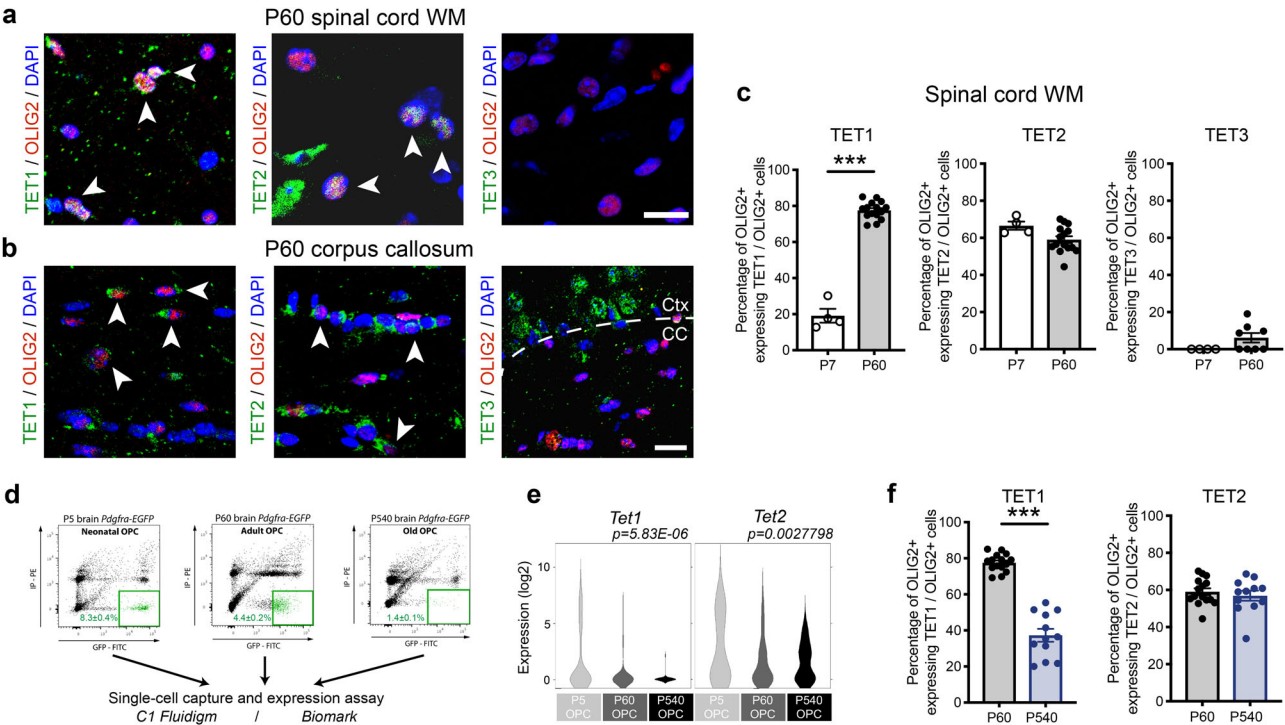

**Fig. 2 TET1 and TET2 are the most abundant isoforms in OLIG2 + cells in the adult spinal cord, with only TET1 displaying an age-dependent decline.**
**a**, **b** Representative confocal images of coronal spinal cord (**a**) and brain (**b**) sections from adult mice (P60) stained with antibodies specific for the indicated TET isoforms (green) and for OLIG2 (in red). White arrowheads indicate co-labeled cells. Scale bar = 50 μm. **c** Percentage of OLIG2 + cells expressing TET1, TET2, or TET3 in postnatal P7 and adult P60 spinal cord. Error bars represent SEM for $n = 4$ (for neonates) and $n = 22$ (for adult) samples (***$p < 0.001$ for TET1, $p = 0.0640$ for TET2, $p = 0.1281$ for TET3, Student's $t$ test two-tailed). **d** Representative fluorescence-activated cell sorting plots of oligodendrocyte progenitor cells isolated from *Pdgfrα-H2BEGFP* reporter mice at neonatal (P5, nOPC), adult (P60, aOPC), and old (P540, oOPC) age. Single-cells were captured and RNA extracted using microfluidics (C1Fluidigm), coupled to RT-qPCR for single-cells, using Biomark. **e** Violin plots of *Tet1* and *Tet2* in nOPC ($n = 61$), aOPC ($n = 76$), and oOPC ($n = 51$) reveal a significant and sharp age-dependent decline of *Tet1* and to a much lesser extent of *Tet2* expression. Data represent log2 expression, derived from Ct (one-way ANOVA, factors age). **f** Bar graphs show the percentage of TET1 or TET2 expressing OLIG2+ cells in young and old spinal cord sections. Average counts in 3–4 sections per mouse for $n = 22$ (young) and $n = 6$ (old) mice, error bars represent SEM (***$p < 0.001$ for TET1, $p = 0.5050$ for TET2, Student's $t$ test two-tailed).

ventral white matter spinal cord sections revealed the same number of myelinated fibers (Fig. 3k, l) and g-ratio (Fig. 3m) in control and *Tet1* mutants. Similar cell density (Fig. 3n, o) and extent of MBP immunoreactivity (Fig. 3p) were detected in the corpus callosum of *Tet1* mutants compared to controls, suggesting no regional differences. *Tet2* mutants did not reveal any difference in myelin content compared to controls (Supplemental Fig. 3e–j).

To further ascertain that lack of *Tet1* did not impact the critical period of developmental myelination, we conducted immunohistochemical analysis of spinal cord sections at earlier developmental time points (postnatal day 14, P14) (Supplementary Fig. 5a). Also, at this time point, the density of OLIG2 + cells and CC1 + oligodendrocytes (Supplementary Fig. 5b) and the extent of immunoreactive MBP areas (Supplementary Fig. 5c) were similar between *Tet1* mutants and controls. Those findings were further validated by electron microscopy on postnatal day 14 spinal cords (Supplementary Fig. 5d, e) and optic nerves (Supplementary Fig. 5g, h) from controls and *Tet1* mutants, which revealed similar numbers of myelinated axons (Supplementary Fig. 5e, h) and similar g-ratio (Supplementary Fig. 5f, i) in mice of the two genotypes. Overall, we conclude that adult myelination was not impacted by the loss of *Tet1* or *Tet2* during development.

To address the functional consequences of *Tet1* ablation on myelin repair, we then induced demyelination by LPC injection in the spinal cord of mice with either constitutive (*Olig1-cre;Tet1-flox*) (Fig. 4a) or tamoxifen-inducible ablation (*Pdgfra-creER(T); Tet1-flox*) of *Tet1* in oligodendrocyte progenitors[42] (Fig. 4n). Constitutive *Tet2* mutants (*Olig1-cre;Tet2-flox*) were used as control for specificity (Fig. 4a). At 14dpl, a lower proportion of OLIG2 + with high 5hmC (Fig. 4b, c), was detected within the area surrounding the lesion in constitutive *Tet1* mutants compared to controls, while no difference was detected in the *Tet2* mutants (Fig. 4d, e). Constitutive *Tet1* mutant mice with defective DNA hydroxymethylation also showed impaired myelin regeneration compared to controls (Fig. 4f, g and Supplementary Fig. 6a), while *Tet2* mutants were virtually indistinguishable from wild-type mice (Fig. 4h, i and Supplementary Fig. 6a). Impaired remyelination in the *Tet1* mutants was validated by electron microscopy (Fig. 4j), which revealed fewer remyelinated axons (Fig. 4k) but unaltered g-ratio (Fig. 4l), compared to controls. The inefficient remyelination in the *Tet1* mutants could not be explained by defective proliferation (Supplementary Fig. 6b) or decreased early differentiation, as the percentage of OLIG2 + and CC1 + cells was the same in mutants and controls at 14dpl (Supplementary Fig. 6c). We therefore reasoned that the changes detected during the repair process were likely consequent to altered gene expression in differentiating CC1 + oligodendrocytes, due to impaired DNA hydroxymethylation (Fig. 4m).

To rule out that the observed phenotype could be linked to any residual effect of *Tet1* loss-of-function during developmental myelination, we repeated these critical experiments in the tamoxifen-inducible lineage-specific *Pdgfra-creER(T);Tet1-flox*

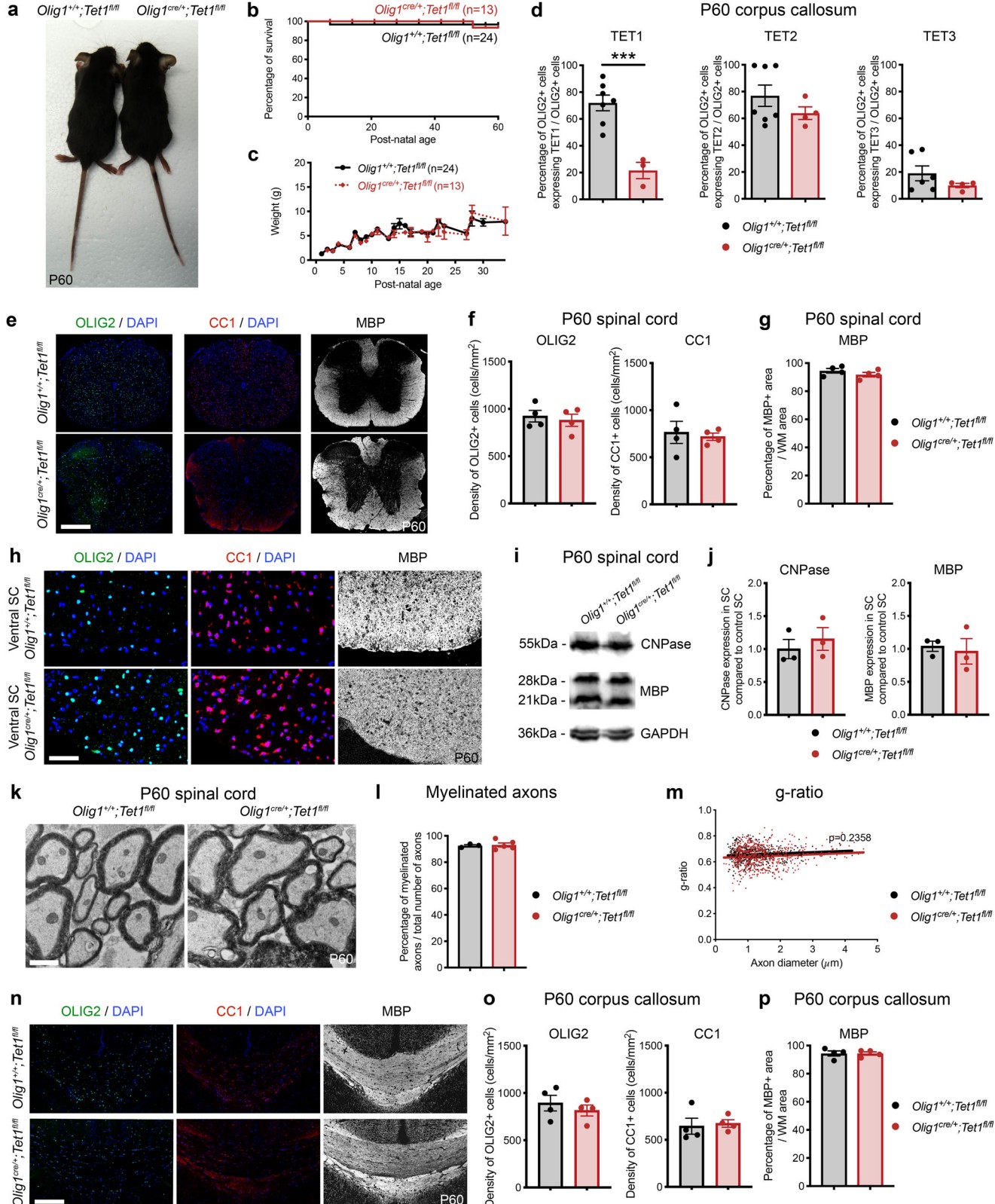

line (Fig. 4n). Adult P60 inducible *Tet1* mutants and control mice received one week of tamoxifen induction by gavage prior to LPC-injection in the spinal cord (Fig. 4n). Immunohistochemistry was used to validate specific recombination, and confirm the lower protein levels of TET1 in OLIG2 + cells—but not in any other cell type—in the inducible *Tet1* mutants compared to controls (Supplementary Fig. 6d). Lower TET1 expression in

OLIG2 + cells was associated with decreased 5hmC immunoreactivity at 14dpl (Fig. 4o), and inefficient remyelination, assessed by Fluoromyelin staining (Fig. 4p) and MBP immunoreactivity in lesioned spinal cords (Supplementary Fig. 6f). Also, in the *Tet1* inducible mutants, the inefficient remyelination could not be attributed to decreased numbers of OLIG2 + and CC1 + cells as the population size was similar in mice of the two genotypes

**Fig. 3 Ablation of *Tet1* during development is compatible with normal myelination in the adult central nervous system. a** Photograph of adult controls (*Olig1*$^{+/+}$;*Tet1*$^{fl/fl}$) and Tet1 mutant (*Olig1*$^{cre/+}$;*Tet1*$^{fl/fl}$) mice reveals no difference in body size. **b** Kaplan–Meier survival curve and **c** body weight for controls ($n = 24$) and *Tet1* mutants ($n = 13$). Error bars represent SEM. **d** Percentage of TET1, TET2, and TET3 expressing OLIG2 cells in the corpus callosum of mice of the two genotypes. Note decreased percentage of OLIG2 + cells expressing TET1 without compensatory increase of TET2 or TET3 expressing cells. Error bars represent SEM for $n = 7$ control mice and $n = 4$ *Tet1* mutants (***$p = 0.0010$ for TET1, $p = 0.2801$ for TET2, $p = 0.2338$ for TET3, Student's $t$ test two-tailed). **e** Representative P60 coronal sections of *Olig1*$^{+/+}$;*Tet1*$^{fl/fl}$ and *Olig1*$^{cre/+}$;*Tet1*$^{fl/fl}$ spinal cords, stained for OLIG2 (green), CC1 (red), MBP (white), DAPI (blue). Scale bars = 500 μm. **f** Quantification of OLIG2 + and CC1 + cell density (number of cells per mm$^2$) in the spinal cord of mice of the indicated genotypes. Error bars represent SEM for $n = 4$ mice ($p = 0.6482$ for OLIG2, $p = 0.7252$ for CC1, Student's $t$ test two-tailed). **g** Average MBP + area relative to total white matter (WM) area in coronal spinal cord sections. Error bars represent SEM for $n = 4$ mice ($p = 0.2693$, Student's $t$ test two-tailed). **h** Representative confocal micrograph of adult ventral spinal cord in coronal sections from mice of the indicated genotype, stained for OLIG2 (green), CC1 (red), MBP (white), DAPI (blue). Scale bars = 100 μm. **i** Representative Western blot of protein extract from control and *Tet1* mutants reveals similar levels of the myelin proteins CNPase, and MBP. GAPDH used as loading control. Molecular weight indicated on the left. **j** Average levels of the myelin proteins CNPase (left) and MBP (right), relative to GAPDH, quantified in three independent experiments. Error bars represent SEM for $n = 3$ spinal cords ($p = 0.5347$ for CNPase, $p = 0.7295$ for MBP, Student's $t$ test two-tailed). **k** Representative ultra-micrograph of adult ventral white matter spinal cords from mice of the indicated genotype revealing no difference in myelination between controls and mutants. Scale bar = 5 μm. **l** Percentage of myelinated axons relative to total axons in sections of P60 *Olig1*$^{+/+}$;*Tet1*$^{fl/fl}$ and *Olig1*$^{cre/+}$;*Tet1*$^{fl/fl}$ white matter spinal cords. Error bars represent SEM for $n = 3$ control mice and $n = 5$ *Tet1* mutants ($p = 0.8009$, Student's $t$ test two-tailed). **m** Scatter plot of the g-ratio in myelinated axons of the indicated caliber, in adult control (black) and *Tet1* mutant (red) spinal cords. A total of $n = 116$ axons in 3 control mice and $n = 199$ axons in 5 *Tet1* mutants were quantified (non-linear regression, $p = 0.2358$, comparison of slopes with sum-of-squares F Test). **n** Representative confocal image of P60 coronal sections from *Olig1*$^{+/+}$;*Tet1*$^{fl/fl}$ and *Olig1*$^{cre/+}$;*Tet1*$^{fl/fl}$ corpus callosum, stained for OLIG2 (green), CC1 (red), MBP (white), DAPI (blue). Scale bar = 500 μm. **o** Quantification of OLIG2 + and CC1 + cell density (number of cells per mm$^2$) in the corpus callosum of mice of the indicated genotypes. Error bars represent SEM for $n = 4$ mice ($p = 0.4644$ for OLIG2, $p = 0.7750$ for CC1, Student's $t$ test two-tailed). **p** Average MBP + area relative to total white matter (WM) area in corpus callosum. Error bars represent SEM for $n = 4$ mice ($p = 0.9960$, Student's $t$ test two-tailed).

(Supplementary Fig. 6e). Collectively, these results suggest that the inability of adult OPCs to reach high levels of DNA hydroxymethylation, as seen in old mice and in the *Tet1* constitutive mutants, did not impair the competence of adult OPCs to either proliferate or start differentiating, although it possibly impaired the expression of other genes regulated by DNA hydroxymethylation and critical for myelin repair.

**DNA hydroxymethylation modulates the expression of genes important for myelin repair.** To define the identity of the genes regulated by DNA hydroxymethylation during myelin repair in the adult CNS, we adopted a combination of experimental strategies using adult reporter mice[43–45] and unbiased DNA and RNA sequencing approaches. The *Pdgfra-EGFP* mice expressing EGFP-tagged histone H2B[44] were used to sort adult OPCs, following a modified protocol for the removal of meninges and blood vessels (Supplementary Fig. 7a)[38]. This allowed the isolation of a 90% pure population of progenitors identified by immunoreactivity for PDGFRα and NG2 (Supplementary Fig. 7b) and expression of OPC-specific transcripts (Supplementary Fig. 7c). The *Plp-EGFP* mice[43,45] were used to isolate adult OLs, using a similar protocol (Supplementary Fig. 7d). The sorted cell population was characterized by 93% of cells with MBP and MOG immunoreactivity (Supplementary Fig. 7e) and enrichment for myelin transcripts (Supplementary Fig. 7f).

The genome-wide distribution of the 5hmC mark was determined by Reduced Representation Hydroxymethylation Profiling (RRHP) of DNA samples from these two characterized populations of sorted adult OPCs and adult OLs. The transcriptional profiles were defined by RNA-sequencing analysis of RNA samples extracted from the same sorted cells (Fig. 5a). RRHP provides an assessment of 5hmC levels at single base resolution, as the method relies on the glucosylation of 5hmC by a specific enzyme, which renders that specific cytosine insensitive to digestion by *MspI* (an enzyme recognizing and digesting the internal cytosine of CCGG sites regardless of methylation) and therefore allows for adapter ligation, amplification, and further sequencing. We verified that an equivalent number of CpGs was present in the DNA samples isolated from aOPC and aOL samples (Supplementary Fig. 8a). We also checked that the

hydroxymethylation coverage was similar among distinct biological replicates, although as expected, higher sequencing reads were detected in samples with greater levels of 5hmC (Supplementary Fig. 8b).

We detected a total of 5583 genes with statistically significant ($q$-value < 0.05) differentially hydroxymethylated regions (DhMR, defined by the presence of at least two concordant CpGs within 2 kb) in aOL compared to aOPC. The overwhelming majority of the genes differentially hydroxymethylated during the transition from adult OPC to OL (5511 out of 5583 genes) was characterized by a 40–60% increase of hydroxymethylation, mostly distributed at promoters and gene regions (Fig. 5b, c). Those hydroxymethylated genes included ontology categories involved in positive regulation of differentiation and cell communication (Fig. 5d). The remaining few genes (72 out of 5583 genes), with decreased hydroxymethylation levels, included those involved in negative regulation of cell communication (Fig. 5e). Overall, these data identified hydroxymethylation as an epigenetic mark regulating the expression of genes involved in communication between myelinating cells and neurons. To further define the transcriptional consequences of hydroxymethylation, we analyzed the RNA-sequencing datasets from the same samples of sorted cells (Fig. 5a) and identified transcripts that were either downregulated ($n = 1187$, Fig. 5f) or upregulated ($n = 1739$, Fig. 5g) in adult OLs compared to adult OPCs ($p$-value < 0.01, $q$-value < 0.05). Among the downregulated gene ontology categories we identified cell migration, proliferation, communication, and cell adhesion (Fig. 5f), while the upregulated GO categories included cell communication and metabolic processes (Fig. 5g). Among the significantly upregulated transcripts involved in regulation of cell communication, we identified the solute carrier family, such as the anion and cation transporters *Slc12a2*, *Slc22a23*, and *Slc9a6* (Fig. 5g). By intersecting the transcriptomic datasets with the list of genes differentially hydroxymethylation at promoters (Fig. 5h) and at gene regions (Fig. 5i), we were able to conclude that 91.7% of the transcripts upregulated in adult OLs compared to adult OPCs were characterized by hydroxymethylation at promoters and 99.0% at gene regions, with several genes showing DNA hydroxymethylation at both genomic locations. We therefore combined the two lists (1595 genes with increased

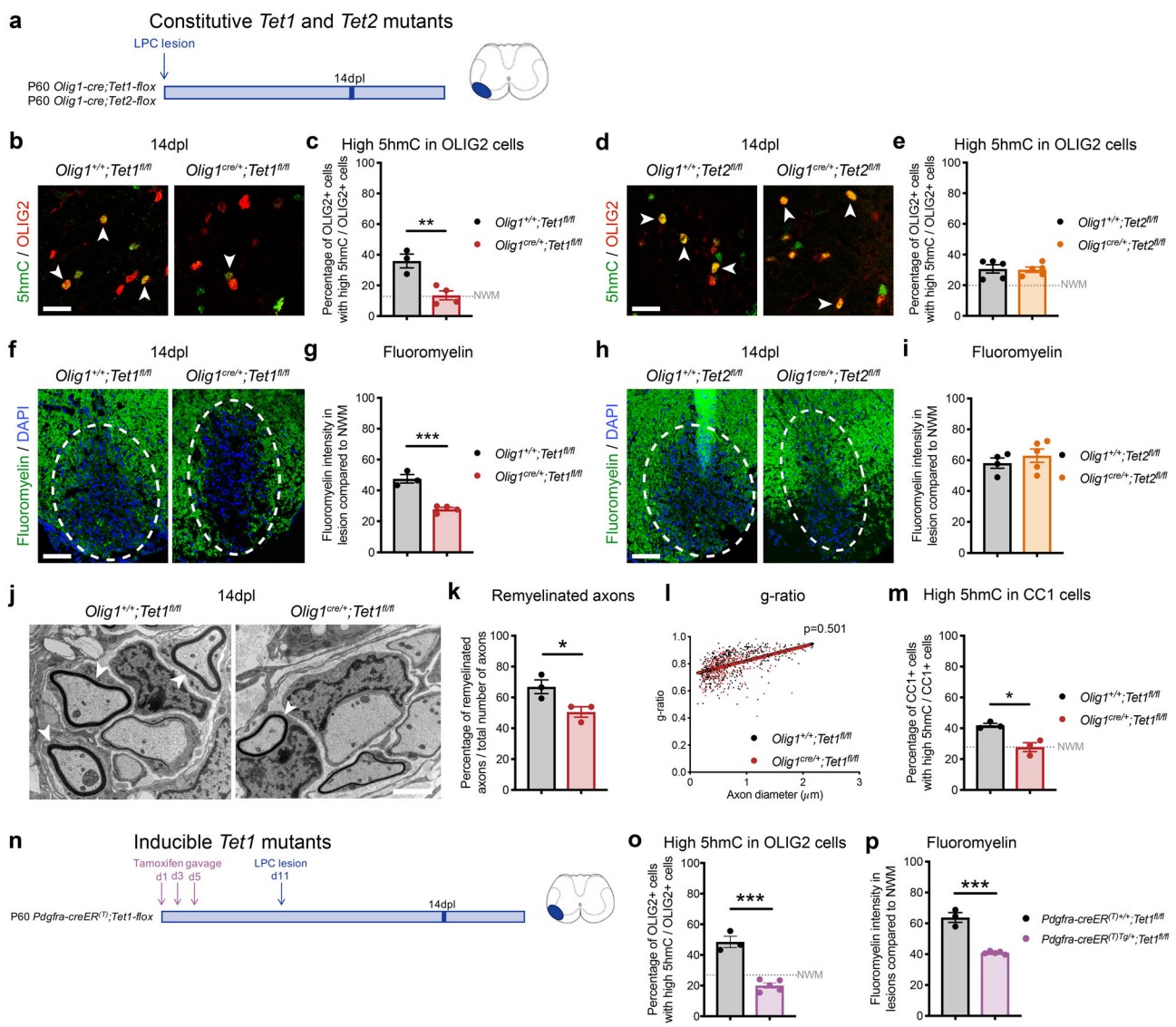

**Fig. 4 Constitutive or inducible ablation of *Tet1* in OPC mimic the inefficient myelin repair detected in old mice. a** Experimental design in constitutive *Tet1* or *Tet2* knockout mice and controls. **b, d** Representative confocal image of spinal cord sections stained for 5hmC (green) and OLIG2 (red) at 14dpl in *Tet1* (**b**) or *Tet2* (**d**) mutant mice. White arrowheads indicate OLIG2 + cells with high levels of 5hmC immunoreactivity. Scale bar = 50 μm. **c, e** Percentage of OLIG2 + cells with high level of 5hmC in lesioned spinal cord sections at 14dpl in *Tet1* (**c**) or *Tet2* (**e**) mutants compared to that in wild-type mice. Dotted line indicates relative percentages in unlesioned white matter in the spinal cord (NWM). Average counts quantified in 3–4 sections/mouse for *n* = 3 control mice, *n* = 4 *Tet1* mutants (**c**) and *n* = 5 for control and *Tet2* mutants (**e**). Error bars represent SEM (**p = 0.0070 for (**c**), *p* = 0.8713 for (**e**), Student's *t* test two-tailed). **f, h** Representative confocal images of spinal cord lesions stained for Fluoromyelin (green) at 14dpl in controls and *Tet1* (**f**) or *Tet2* (**h**) mutants (lesion area as white dashed line). Scale bar = 100 μm. **g, i** Fluoromyelin average intensity in control (gray bars) and *Tet1* (red bar in **g**) or *Tet2* (orange bar in **i**) lesions relative to the levels in unlesioned white matter (NWM) calculated for *n* = 3 control mice, *n* = 4 *Tet1* mutants (**g**) and *n* = 5 for control and *Tet2* mutants (**i**). Error bars represent SEM (***p = 0.0007 for **g**, *p* = 0.4304 for **i**, Student's *t* test two-tailed). **j** Electron micrographs of remyelination (remyelinated axons indicated by white arrowheads) at 14dpl in *Olig1^{+/+};Tet1^{fl/fl}* and *Olig1^{cre/+};Tet1^{fl/fl}* spinal cords. Scale bar = 5 μm. **k** Percentage of remyelinated axons in controls (gray) and *Tet1* mutants (red) at 14dpl, relative to total number of axons. Error bars represent SEM for *n* = 3 mice (*p = 0.0433, Student's *t* test two-tailed). **l** Scatter plot of the g-ratio in myelinated axons of the indicated caliber in the lesioned spinal cord of wild-type mice (black) and *Tet1* mutants (red) at 14dpl. A total of 107–150 axons were quantified for each mouse and *n* = 3 mice for each genotype were assessed (non-linear regression *p* = 0.501, comparison of slopes with sum-of-squares F Test). **m** Percentage of CC1 + cells with high level of 5hmC in lesioned spinal cord evaluated at 14dpl in mice of both genotypes. Dotted line indicates the percentage of CC1 + cells with high level of 5hmC in unlesioned tracts. Average counts from 3 quantified sections/mouse. Error bars represent SEM for *n* = 3 mice (*p = 0.0109, Student's *t* test two-tailed). **n** Experimental design in inducible P60 *Pdgfra-creER^{(T)+/+};Tet1^{fl/fl}* and *Pdgfra-creER^{(T)Tg/+};Tet1^{fl/fl}* mice, after tamoxifen induction. **o** Percentage of OLIG2 + cells with high levels of 5hmC at 14dpl in *Pdgfra-creER^{(T)+/+};Tet1^{fl/fl}* and *Pdgfra-creER^{(T)Tg/+};Tet1^{fl/fl}* spinal cords, after tamoxifen induction. Dotted line indicates values in unlesioned tracts. Error bars represent SEM for *n* = 3 control mice and *n* = 5 inducible *Tet1* mutants (***p = 0.0002, Student's *t* test two-tailed). **p** Quantification of Fluoromyelin intensity in lesioned spinal cord at 14dpl, compared to normal white matter (NWM), in *Pdgfra-creER^{(T)+/+};Tet1^{fl/fl}* and *Pdgfra-creER^{(T)Tg/+};Tet1^{fl/fl}* after tamoxifen induction. Data represent the average intensity in lesioned compared to unlesioned areas. Error bars represent SEM for *n* = 3 control mice and *n* = 5 inducible *Tet1* mutants (***p < 0.001, Student's *t* test two-tailed).

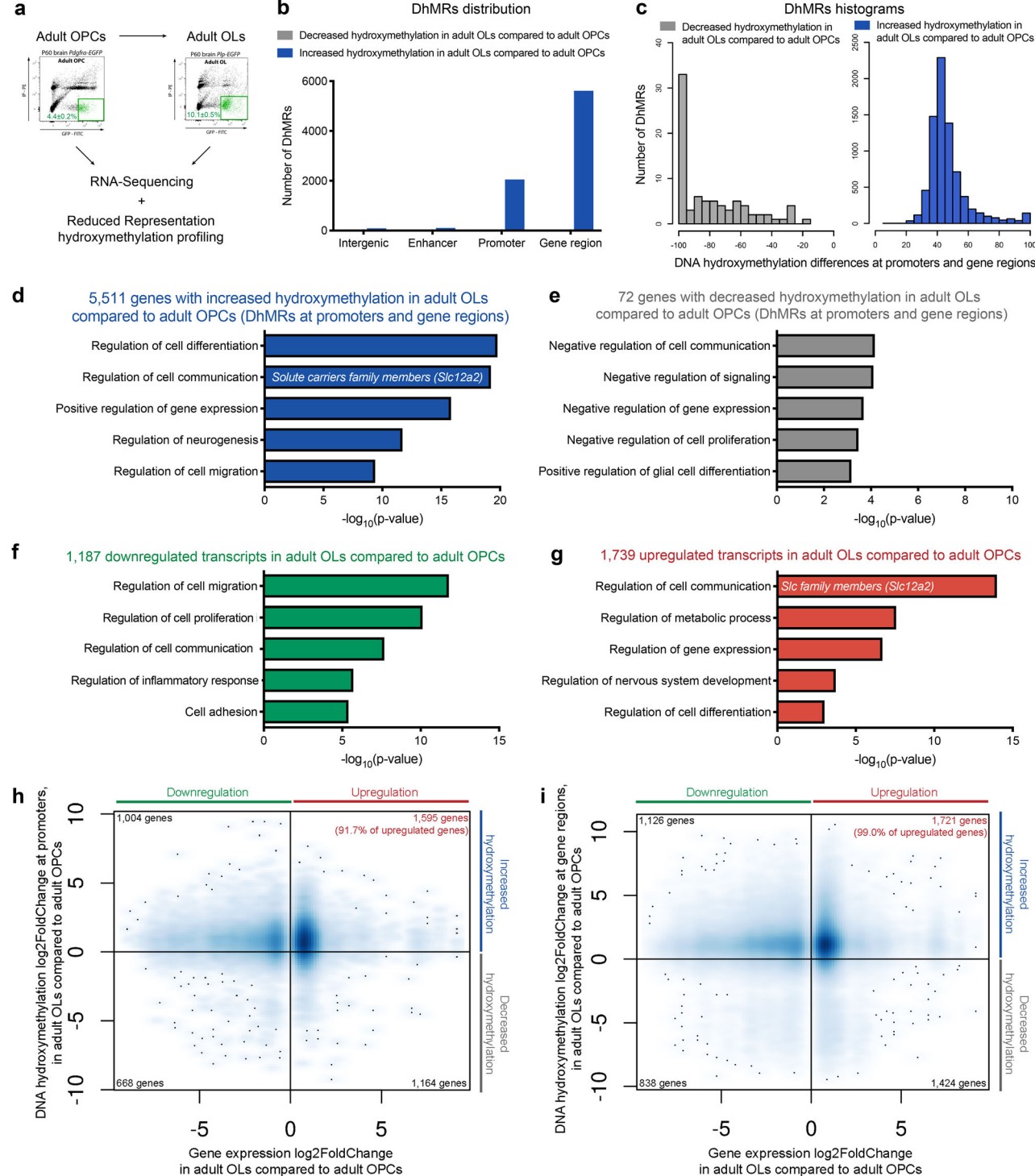

hydroxymethylation at promoter regions and 1721 at gene body regions) to obtain a total of 1726 unique genes with increased expression and increased hydroxymethylation in adult OLs, compared to OPCs. Among those genes we identified the solute carrier gene family (Fig. 5h, i and Supplementary Table 1). As the distribution of hydroxymethylation at gene body regions raised the potential issue of a bias of this DNA modification for larger genes, we conducted a correlation analysis between the number of hydroxymethylated CpGs and gene size (Supplementary Fig. 8c). This analysis revealed that the hyper-hydroxymethylation detected in adult OLs, in particular for the solute carrier gene family, was specific to the stage of differentiation of the cells and not of the

result of large gene size. It is also worth noting that this DNA modification was not uniquely linked to transcriptional activation, as 1142 genes with increased DNA hydroxymethylation (1004 at promoters and 1126 at gene regions) were characterized by lower transcript levels in adult OLs compared to adult OPCs (Fig. 5h, i), suggesting a more global role for DNA hydroxymethylation on gene regulation.

To better understand the relationship between TET1-dependent DNA hydroxymethylation and gene expression in adult OLs during the remyelination process, we then compared these lists of unique genes with the differential transcriptomic dataset obtained from preparation of spinal cord samples isolated

**Fig. 5 DNA hydroxymethylation and gene expression in sorted adult oligodendrocyte progenitors and adult oligodendrocytes. a** Flow-activated cell-sorting of adult OPCs from *Pdgfra-H2BEGFP* and adult OLs from *Plp-EGFP* mice followed by RNA and DNA extraction and processing for RNA-Sequencing and RRHP DNA hydroxymethylation analysis. **b** Gene features distribution of differentially hydroxymethylated (DhMR) regions in aOLs relative to aOPCs. The bar graphs indicate the number of regions with increased (blue) or decreased (gray) hydroxymethylation at intergenic, enhancer, promoter, and gene regions. **c** Histograms of the overall distribution of hypo- (gray) and hyper- (blue) hydroxymethylated regions between aOLs and aOPCs samples. The bar graphs identify the number of DhMR at each level of decile DNA hydroxymethylation difference. **d** Ontology categories of the 5511 genes with increased hydroxymethylation, corresponding to the following GO, in decreasing order (from top to bottom: GO:0030154, GO:0010646, GO:0010628, GO:0050767, GO:0030334 (blue). **e** Only 72 genes showed decreased hydroxymethylation (gray), in aOLs compared to aOPCs, as defined by the RRHP analysis. **f** Gene ontology categories of the 1187 transcripts downregulated (green) in aOLs compared to aOPCs. From top to bottom, GO:0030334, GO:0042127, GO:0010646, GO:0050727, GO:0007155. **g** Gene ontology categories of the 1739 transcripts upregulated (red) in aOLs compared to aOPCs, as identified by RNA Sequencing analysis. From top to bottom, GO:0050794, GO:0019222, GO:0010468, GO:0051960, GO:0045595. **h**, **i** Quadrant plot of the relationship between genes with differential hydroxymethylation at promoters (**h**) or gene regions (**i**) and differential transcript (aOLs vs. aOPCs). The *x* axis indicates log2 fold change of transcript levels. The *y* axis refers to DNA hydroxymethylation differences. Note the "hot spot" (blue cloud of dots) identifying the abundance of genes with increased hydroxymethylation at promoters and gene regions associated with upregulated transcripts.

four days post-LPC (4dpl) injection of either wild-type control mice (*Olig1*$^{+/+}$;*Tet1*$^{fl/fl}$) or *Tet1* mutants (*Olig1*$^{cre/+}$;*Tet1*$^{fl/fl}$) (Fig. 6a). Samples isolated from the unlesioned contralateral side of the spinal cords, were used as additional control, in both wild-type mice and *Tet1* mutants (Fig. 6a). Consistent with the previously described normal levels of adult myelin, the transcriptome of unlesioned spinal cord samples from mice of the two genotypes, was remarkably similar, with only 7 transcripts being differentially expressed (*p* value < 0.01, *q* value < 0.05). In contrast, the transcriptome of spinal cord samples isolated during the myelin repair phase, was dramatically different between the two genotypes. Preparations obtained from wild-type mice at 4dpl were characterized by a substantial reorganization of gene expression during myelin repair, with 2469 upregulated and 1609 downregulated transcripts compared to unlesioned tissue (*p* value < 0.01, *q* value < 0.05). Preparations obtained from *Tet1* mutants after lesion, in contrast, were characterized by much smaller changes, with only 121 upregulated and 498 downregulated transcripts (*p* value < 0.01, *q* value < 0.05), compared to unlesioned tissue. The fact that 2369 (95.9%) of the 2469 transcripts with increased expression after LPC injection in wild-type spinal were not detected in the *Tet1* mutants, clearly suggested a blunted transcriptional response during myelin repair in the absence of DNA hydroxymethylation (Fig. 6b). Those transcripts included several gene categories related to lipid metabolism (GO:008160), inflammation, autophagy, glial cell activation, cell migration, cellular homeostasis, and response to stress (Fig. 6b). Since hydroxymethylation regulates the expression of genes needed for adult myelin formation, we then asked whether those genes overlapped with the ones regulated by increased hydroxymethylation during the differentiation of adult OPCs into adult OLs (Fig. 5h–i). This analysis detected a statistically significant overlap in 442 genes (*p* = 2.202774e$^{-33}$, Fisher test), which we identified as TET1-gene targets, regulated by DNA hydroxymethylation during myelin repair (Fig. 6d). Among them we found several genes encoding for anion and cation transporters, such as those belonging to the solute carrier family. As predicted, a similar analysis for the genes that were downregulated in wild type but not in *Tet1* mutants (Fig. 6c) did not reveal a statistically significant overlap (*n* = 52 genes, *p* = 1, Fisher test) (Fig. 6e). These results define TET1 function as dispensable for developmental myelination, but essential for the activation of a transcriptional program associated with myelin repair after demyelination.

**The TET1-target gene SLC12A2 is localized at the axon–myelin interface**. As a proof-of-principle, we decided to focus on a TET1-target gene and selected one of the members of the solute

carrier gene family regulating anion and cation transport (Fig. 5d, g, Supplementary Table 1). We opted for the sodium/potassium/chloride symporter SLC12A2 (also known as NKCC1), since it was previously reported to regulate the response of oligodendrocyte lineage cells to GABAergic signals[46] and to be enriched in newly formed OLs in four distinct transcriptomic datasets[47], including those from OL directly sorted from the brain[48–51], and in proteomic datasets from mature oligodendrocytes[51]. The enrichment of this transcript and protein in oligodendroglial cells urged us to further characterize its expression and regulation by TET1.

To verify that *Slc12a2* expression was indeed regulated by TET1, we induced the overexpression of the human TET1 catalytic domain (hTET1-CD) in vitro on immunopanned oligodendroglial cultures (Fig. 7a, b). TET1 overexpression was sufficient to induce upregulation of *Slc12a2* transcripts in cultured cells (Fig. 7b, c), confirming that *Slc12a2* is a TET1-target gene.

SLC12A2 protein expression was first assessed by immunocytochemistry in mixed glial and neuronal cultures, where it was detected at low levels as differentiating OPCs contacted neuronal processes (Fig. 7d). We then assessed SLC12A2 expression in vivo, using immuno-gold electron microscopy of adult optic nerve sections, which unequivocally localized this transporter at the axon–myelin interface, within the first myelin wraps in close proximity to the axon (Fig. 7e). Single-cell rt-qPCR of oligodendrocyte lineage cells sorted from neonatal (P5), young adult (P60), and old (P540) brain, further confirmed that the *Slc12a2* transcripts followed the same age-related decline as *Tet1* (Fig. 7f). Consistent with SLC12A2 being the product of one of the TET1-target genes, its protein expression also declined in the spinal cord of older mice (Fig. 7g). The time course of SLC12A2 expression during myelin repair in young mice was consistent with the temporal occurrence of DNA hydroxymethylation and the generation of new OL, whereas the inefficient remyelination in older mice was associated with reduced DNA hydroxymethylation and decreased SLC12A2 immunoreactivity (Fig. 7g, h). The localization of SLC12A2 at the axo-myelin interface in young adult mice (Fig. 7e) and the detection of swellings at the axon–myelin interface in older wild type mice (Fig. 7i, j), suggested the importance of TET1-target genes in regulating fluid accumulation at this interface.

**Ablation of the TET1-target gene *Slc12a2* in zebrafish induces swellings in myelinated axons, reminiscent of those detected in old wild type and young *Tet1* mutant mice**. As expected from a TET1-target gene, the levels of SLC12A2—as measured by using immuno-gold EM—were reduced in the optic nerve of *Tet1* mutants compared to wild type (Fig. 8a, b). We also noted that

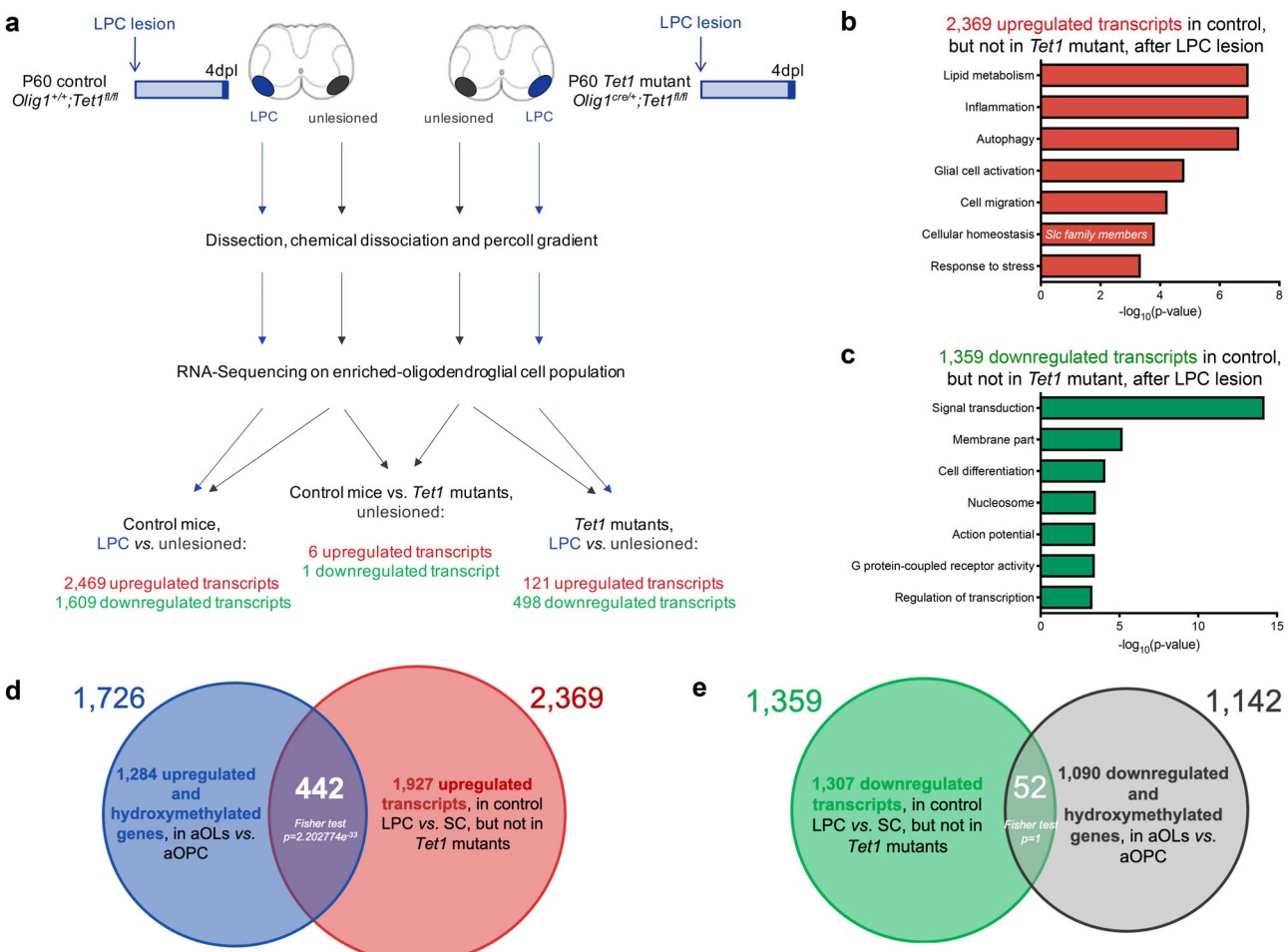

**Fig. 6 Identification of TET1-target genes during new myelin formation in adult CNS. a** Schematic of the experimental design to identify TET1 target genes during myelin repair in young adult mice. Control ($Olig1^{+/+};Tet1^{fl/fl}$) and Tet1 mutant ($Olig1^{cre/+};Tet1^{fl/fl}$) mice were injected with lysolecithin. At 4 day-post-lesion (4dpl), the demyelinated region and contralateral unlesioned tissue were used for oligodendroglial enriched preparation followed by RNA-sequencing. Arrows identify the comparisons that were performed to identify differential gene expression among samples. Myelin repair in control mice was associated with 2469 up- (red) and 1609 down- (green) regulated genes in lesioned tissue compared to unlesioned. In Tet1 mutants, 121 transcripts were up- (red) and 498 down- (green) regulated. Unlesioned tissue of controls and Tet1 mutants showed a total of 7 genes as differentially expressed (6 up- and 1 downregulated). **b** Ontology categories of the 2369 genes that were upregulated during myelin repair only in control spinal cord, but not in Tet1 mutants. Corresponding GO numbers listed in order from top to bottom: GO:0044242, GO:0032675, GO:0006914, GO:0061900, GO:0030335, GO:0019725, GO:0006950. **c** Ontology categories of the 1359 genes that were downregulated during myelin repair only in control spinal cords, but not in Tet1 mutants. Corresponding GO numbers listed in order from top to bottom: GO:0007165, GO:0021515, GO:0001508, GO:0003257, GO:0008528, GO:0044459, GO:0000786. **d** Venn diagram comparing genes upregulated and with increased hydroxymethylation (at promoters and gene regions) in aOLs vs. aOPCs (blue circle) with genes upregulated in lesioned vs. unlesioned spinal cord in control but not in Tet1 mutants (red). Note the significant overlap ($p = 2.202774e^{-33}$, Fisher test two-tailed), of 442 common genes with increased hydroxymethylation and expression levels in newly formed aOLs. **e** Venn diagram comparing genes downregulated and with increased hydroxymethylation (at promoters and gene regions) in aOLs vs. aOPCs (gray) with genes downregulated in lesioned vs. unlesioned spinal cord in control but not in Tet1 mutants (green). The overlap of 52 genes was not statistically significant ($p = 1$, Fisher test two-tailed).

the increased SLC12A2 immunoreactivity detected in wild type during myelin repair after demyelination, was not detected either in constitutive (Fig. 8c, d) or inducible (Supplementary Fig. 6g) Tet1 mutants. We also showed that the lower levels of *Slc12a2* in Tet1 mutants, were associated with significantly enlarged and swollen space at the axon–myelin interface (Fig. 8e, f), a feature that was reminiscent of the swollen axo-myelinic interface detected in old mice (Fig. 7i, j).

To further define a relationship between swelling at the axon–myelin interface and decreased SLC12A2 levels, we used the zebrafish model system for additional loss-of-function studies. Zebrafish mutants, characterized by a loss of function mutation (*ue58*) in the gene encoding for a homologue of *Slc12a2*

(*slc12a2b*), were identified in an ENU-based forward genetic screen[52]. Myelin formation was first assessed using the stable transgenic reporter Tg(mbp:EGFP-CAAX) in *slc12a2b*$^{ue58}$ mutants and compared it to sibling controls. While myelin formed normally in mutants, we noted the appearance of abnormal fluorescent profiles, indicative of swellings around myelinated axons in the CNS (Fig. 8g). To further characterize this pathology, we carried out time-course analyses of single oligodendrocytes labeled with mbp:mem-Scarlet, which allowed to assess the cellular morphology in better detail (Fig. 8h, i). This analysis showed that although *slc12a2b*$^{ue58}$ mutant and control cells looked similar at early stages of myelin formation (4 day-post fertilization, 4dpf) (Fig. 8h), a clear pathology, characterized

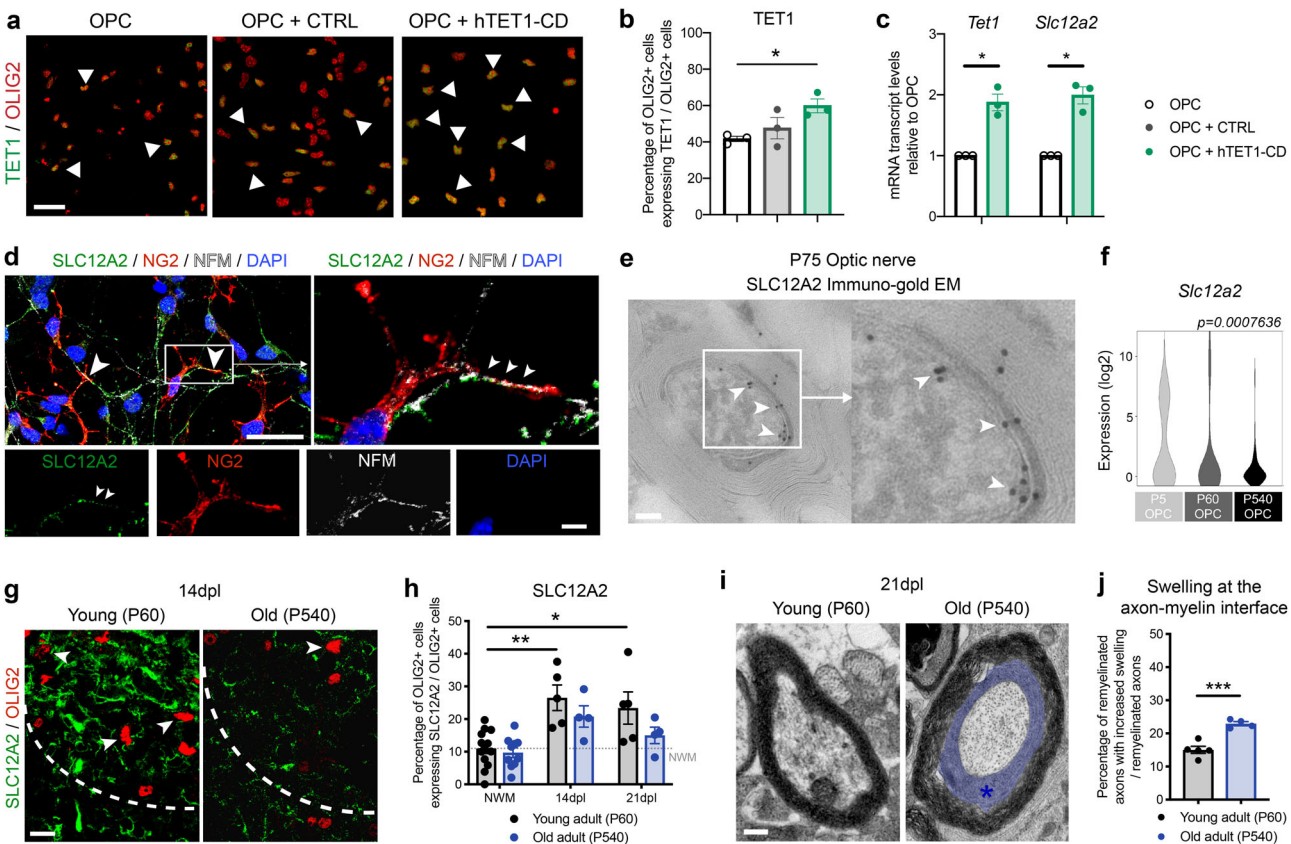

**Fig. 7 The protein encoded by the TET1-target gene Slc12a2 is localized at the axon–myelin interface. a** Representative images of cultured OPCs, transduced with the indicated lentiviral vectors and then stained for TET1 (green) and OLIG2 (red). White arrowheads indicate co-labeled cells. Scale bar = 25 µm. **b** Quantification of the percentage of OLIG2 + cells expressing TET1 in untransduced OPC and in cells transduced either with CTRL or hTET1-CD lentiviral vectors. Error bars represent SEM in $n = 3$ independent experiments (*$p < 0.05$, one-way ANOVA). **c** Quantitative real-time PCR analysis of *Tet1* and *Slc12a2* transcript levels in immunopanned OPCs after transfection with hTET1-CD lentivirus. Data represent average transcript levels relative to untransfected OPCs in $n = 3$ independent experiments. Error bars represent SEM (*$p = 0.0232$ for *Tet1*, *$p = 0.0192$ for *Slc12a2*, Student's *t* test two-tailed). **d** Confocal image of mixed neuronal/glial cortical cultures. SLC12A2 (in green) is expressed by differentiating oligodendrocyte progenitors (co-stained with NG2, in red), and localized at contact points with axons (stained with NFM, in white). DAPI (blue) used as nuclear counterstain. White arrowheads indicate NFM + and NG2 + neuron-glial contact points co-labeled with SLC12A2. Scale bars = 50 µm and scale bars = 10 µm for high magnification images. **e** Immunogold staining for SLC12A2 on EM imaging in adult P75 WT optic nerves, showing the localization of SLC12A2 at the axon–myelin interface (white arrow-heads). Scale bar = 5 µm and scale bar = 1 µm for high magnification images. **f** Violin plots of *Slc12a2* transcript levels in nOPC at P5 ($n = 61$ single cells), aOPC at P60 ($n = 76$ single cells), and oOPC at P540 ($n = 51$ single cells). Note the age-dependent decline of *Slc12a2* transcripts (one-way ANOVA, for age). **g** Representative confocal image of young and old spinal cord sections, stained 14 days after lesion for SLC12A2 (green) and OLIG2 (red). The dotted white line indicates the lesion border. White arrowheads indicate co-labeled cells. Scale bar = 50 µm. **h** Quantification of in the percentage of OLIG2 + cells expressing also SLC12A2 in young and old lesioned spinal cords at 14dpl and 21dpl. Dotted gray line across indicates the percentage of OLIG2 + cells expressing SLC12A2 in unlesioned white matter tracts (NWM). Data represent average values quantified in 4 sections/mouse for $n = 5$ young and $n = 4$ old mice (*$p < 0.05$ and **$p < 0.01$, two-way ANOVA, for age and time after lesion). **i** Representative electron micrographs of young and old spinal cord sections at 21dpl, revealing increased swelling at the axo-myelinic interface (blue area and * asterisk). Scale bar = 10 µm. **j** Quantification of the percentage of remyelinated axons with swelling at the neuro-glial interface in young and old spinal cord lesions at 14dpl. Data represent the average values. Error bars represent SEM for $n = 5$ young and $n = 4$ old mice (***$p = 0.0005$, Student's *t* test two-tailed).

by the appearance of localized swellings associated with individual myelin sheaths, emerged over time in *slc12a2b^ue58* mutant cells (Fig. 8i). These results provided further evidence in support of the role of solute carriers in the regulation of the axon–myelin interface.

Taken together, our data support DNA hydroxymethylation as an epigenetic modification critical for proper myelin repair in the CNS of young adult mice, which declines with aging. TET1 is identified as the specific enzyme responsible for this process, as lower TET1 levels, detected with aging or after genetic ablation, result in lower expression of its target genes and impaired myelin repair. Importantly, TET1 function is not essential for the "default" myelination program, as white matter tracts in the adult central nervous system of mutants and wild-type mice are virtually

indistinguishable. In contrast, TET1 function regulated the expression of several genes involved in myelin repair, including the expression of solute carriers, which regulate ion transport at the axon–myelin interface. Thus, the age-dependent decline of TET1 in mice, results in an overall decreased transcriptional program, which prevents efficient repair of demyelinated lesions.

## Discussion

Myelin formation is a well-studied process in the developing central nervous system. Neonatal OPCs globally reorganize their transcriptome to repress genes involved in the regulation of proliferation and migration, as well as transcriptional inhibitors of myelin genes[5,6]. We and others reported on the importance of this developmental process and identified several epigenomic

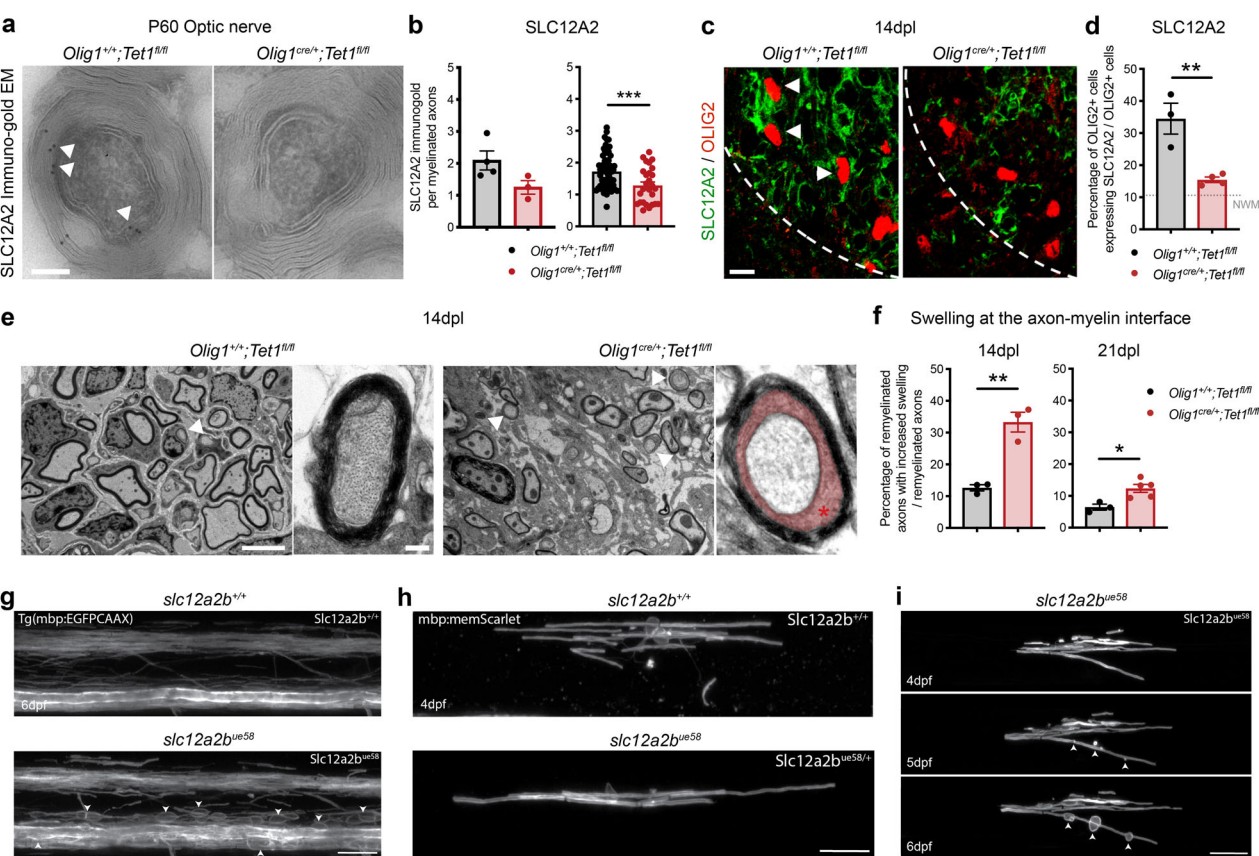

**Fig. 8 Ablation of *Slc12a2b* in zebrafish induces myelin swellings reminiscent of those detected in old mice. a** Ultra-micrograph of immunogold staining for SLC12A2 in P60 *Olig1*[+/+]*;Tet1*[fl/fl] and *Olig1*[cre/+]*;Tet1*[fl/fl] optic nerves SLC12A2 immunoreactive particles localized at the axon–myelinic interface in control tissue(white arrowheads). Scale bar = 5 μm. **b** Quantification of the SLC12A2 particles in optic nerves. Data represent of the average number of SLC12A2 gold particles per mouse (left) and per axon (right). Error bars represent SEM for myelinated axons, quantified in *n* = 4 control and *n* = 3 mutant mice (total of at least 27 and at maximum 64 myelinated axons quantified in one replicate) (*p* = 0.0866, ***p* = 0.0008, Student's *t* test two-tailed).
**c** Representative confocal image of spinal cord at 14dpl stained for SLC12A2 (green) andOLIG2 + (red), in mice of the indicated genotype. White arrowheads indicate co-labeled cells. Scale bar = 50 μm. **d** Percentage of OLIG2 + cells expressing SLC12A2 in lesioned spinal cords in mice of the two genotypes, at the 14dpl time point. Dotted line refers to the percentage of OLIG2 + cells expressing SLC12A2 in unlesioned tracts. Error bars represent SEM for *n* = 3 control mice and *n* = 4 *Tet1* mutants (**p* = 0.0059, Student's *t* test two-tailed). **e** Representative electron micrographs of spinal cord sections from wild type and mutants at 14dpl. Low and high magnification (inset) are shown for *Olig1*[+/+]*;Tet1*[fl/fl] and *Olig1*[cre/+]*;Tet1*[fl/fl]. Note the swelling of the axomyelinic space in *Tet1* mutants (red area and asterisk). Scale bar = 10 μm. **f** Quantification of the percentage of remyelinated axons with increased swelling in *Olig1*[+/+]*;Tet1*[fl/fl] and *Olig1*[cre/+]*;Tet1*[fl/fl] lesions at 14dpl and 21dpl. Data are average counts. Error bars represent SEM for *n* = 3 control mice and *Tet1* mutants at 14dpl, *n* = 3 control mice and *n* = 5 *Tet1* mutants at 21dpl (**p* = 0.0032 at 14dpl, *p* = 0.0164 at 21dpl, Student's *t* test two-tailed). **g** Lateral views of the spinal cord in 6dpf Tg(mbp:EGFP-CAAX) zebrafish where myelinating glia in the CNS are labeled. Compared to wild-type sibling controls (left panel), larvae homozygous for the *slc12a2b*[ue58] mutation (right panel) show disrupted myelin morphology (white arrowheads). Scale bars = 20 μm. **h** Single mbp:mem Scarlet-expressing oligodendrocytes in wild-type (left panel) and *slc12a2b*[ue58] heterozygote siblings (right panel) at 4dpf. Scale bars = 20 μm. **i** Example time course showing the development of the myelin phenotype in single mbp:mem Scarlet-expressing oligodendrocytes in *slc12a2b*[ue58] homozygous mutants from 4 to 6dpf. White arrowheads point to localized areas of myelin disruption. Scale bar = 20 μm.

regulators responsible for this transition, including histone deacetylases[9,18,53], repressive histone methyltransferases[8,54], and DNA methyltransferases[16], which repress gene expression and favor a model of nOPC differentiation due to de-repression of myelin gene expression.

In the adult CNS, aOPCs retain the ability to form new myelin in response to damage or loss, a process that becomes less efficient with aging[1,17,18]. Several groups have attempted to understand the underlying causes for this age-dependent decline in myelin repair. External factors, such as declining growth factor concentrations or changes in stiffness of the extracellular matrix, for instance, have been suggested as potential culprits for the age-dependent decline in remyelination[1,2,33–37]. However, it has been long recognized that nOPC and aOPC are also characterized by a different ability to proliferate, migrate, and differentiate[38,55–61].

For example, aOPCs show decreased responsiveness to the same concentration of growth factors and cytokines compared to nOPCs[62]. From a molecular standpoint, these results support intrinsic differences in the properties of nOPC and aOPC, a concept which is supported by the transcriptional heterogeneity of OPCs at different ages[2,49,51]. Despite the large number of studies supporting age-dependent differences in OPC, there is still relatively little information regarding the epigenetic landscape of aOPC, especially during the age-dependent decline observed in myelin repair.

As DNA methylation is the single most significant epigenetic mark predictive of biological age[20], and yet ablation of the DNMTs in oligodendroglial cells only modestly impacted myelin repair[19], we focused this study on DNA hydroxymethylation, catalyzed by the TET enzymes. These enzymes are methyl-

cytosine dioxygenases, which hydroxylate methylated cytosine residues (5mC) into hydroxylated forms (5hmC) that can be then converted into formyl-cytosine and carboxyl-cytosine. These marks can eventually be excised from the DNA[22], thereby removing the DNA methylation-dependent repression of genes and favoring gene expression[21,22,24–27]. We previously reported that DNA methylation, mediated by DNMT1 is necessary for the transcriptional repression of several inhibitors during the differentiation of nOPCs in the developing brain[16]. Here, we show that higher levels of both DNA methylation and hydroxymethylation are detected in aOLs compared to aOPCs.

The overall genome-wide distribution of hydroxymethylation in aOLs identified in our study, is consistent with previous reports[63–65], as it is primarily localized at promoters and gene body regions. The high levels of hydroxymethylation at gene body is not correlated with greater expression of larger genes in aOLs. It is interesting to note that increased hydroxymethylation in aOLs is correlated with gene up- and downregulation, suggesting a global role of DNA modifications in gene expression, which could be explained by different genomic location and/or abundance at specific loci. Increased hydroxymethylation is detected at genes regulating cell–cell communication and lipid metabolism, rather than myelin genes, a finding which is consistent with the lack of a developmental myelin phenotype in *Tet1* mutants. Together, these data support the unique functional importance of DNA hydroxymethylation in newly formed OLs, to coordinate the transcriptional activation of genes regulating lipid metabolism and genes regulating ion homeostasis at the axon–myelin interface. As such, successful myelin repair in the spinal cord of young mice is preceded by increased levels of hydroxymethylation as aOPCs transition into newly formed OLs, which express genes involved in lipid metabolism and ion exchange at the axon–myelin interface. In contrast, age-dependent decline of hydroxymethylation results in inappropriate levels of gene expression in old mice and inefficient myelin repair, characterized by the presence of aberrant swellings at the axon–myelin interface.

Our data also identify TET1 as the main enzyme catalyzing DNA hydroxymethylation in oligodendroglial lineage cells in the young adult spinal cord and whose levels decline with age. TET2 is also expressed in aOPCs, but its levels remain constant over time. TET3, although previously reported in neonatal cultures of cortical oligodendrocyte lineage cells[28], is not detectable in OPCs or OLs in the adult spinal cord. Reduced TET1 levels in older mice are associated with reduced hydroxymethylation in oligodendroglial cells, a finding detected also in young adult mice with oligodendrocyte lineage-specific ablation of *Tet1*, but not of *Tet2*. This underscores the importance of TET1 as the main enzyme responsible for the deposition of the 5hmC epigenetic mark in aOPCs as they differentiate into aOL, and raises the question on functional differences between the TET1 and TET2 isoforms. A potential explanation for the isoform specific role of TET1 in aOPCs, could be presence of a "CXXC" domain, which enables the protein to directly bind DNA and is not present in TET2[24,66]. It is also conceivable that TET1 and TET2 may bind to distinct protein partners in the nuclei of newly formed aOLs and in turn activate TET1 and inhibit TET2 enzymatic activity[67], a question that deserves future investigation.

The higher levels of hydroxymethylation detected in aOPCs compared to nOPCs, are consistent with the concept that distinct epigenetic marks regulate myelin formation in the developing and adult CNS. Developmental myelination is characterized by the establishment of a repressive epigenomic landscape as nOPCs differentiate into nOLs. As such, it is not surprising that neither *Tet1* nor *Tet2* mutants show defective developmental myelination. However, it is important for aOPCs to activate transcription using TET1-dependent hydroxymethylation as cells transition

from a progenitor stage to newly formed aOLs. This modification does not impact the process of differentiation from aOPCs to aOLs, but rather impacts the expression of genes that regulate the stability of the myelin sheath, including genes regulating lipid metabolism and regulate ion homeostasis and cell communication. Among the most significant hydroxymethylated genes whose transcripts are also upregulated during myelin repair, we identify the family of solute carriers, which includes the *Slc12a2* gene. The protein encoded by this gene—also known as NKCC1—is a sodium/potassium/chloride transporter, previously reported to be expressed in neuronal[68,69] and glial cells[46,68,69]. Since SLC12A2/NKCC1 regulates the overall intracellular chloride concentration in neurons and in nOPCs, it has been reported to impact the response of OPC to GABAergic inputs[46,69]. At the same time, due to its regulation of cation concentration, this molecule is of particular importance in regulating also calcium influx, especially in pathological conditions affecting the developing brain, such as hypoxia/ischemia[70,71]. Besides its role as ion co-transporter, SLC12A2/NKCC1 is also highly expressed in epithelial cells, where it is strongly polarized and confined to the basolateral membrane[70,71]. Oligodendrocytes are also highly polarized cells and recent transcriptomic and proteomic studies support the highest levels of SLC12A2 in newly formed myelinating oligodendrocytes[47–51]. Our results confirm these most recent findings, as we detect high levels of SLC12A2 at the axon–myelin interface during myelin repair in young, but not in old mice. As SLC12A2 has been shown to regulate cell volume in epithelial cells[72,73] and promote cell shrinkage in the mammalian brain[74,75], we hypothesize that its localization within the first myelin wraps close to the axon, is indicative of a regulatory role of the volume of the axo-myelinic space. Consistent with this hypothesis, we detect swelling of this space in the CNS of zebrafish mutants for *slc12a2b*, a finding which was also detected in peripheral axons[52]. It is worth noting that this phenotype is distinct from the one previously reported for the *slc12a2a* mutant, which was described to have small ears, due to the loss of endolymph, but no myelin phenotype either in the PNS or in the CNS[76]. Overall, these data suggest that *slc12a2b* can compensate for the *slc12a2* loss, while *slc12a2* is not sufficient to compensate the *slc12a2b* zebrafish mutation. The potential impact of the axo-myelinic swelling detected in mice and zebrafish with decreased expression of TET1 or its gene target *Slc12a2*, has been highlighted by a recent report that changes in the space surrounding the axon, may decrease the speed of axial conduction of the action potential[77]. Collectively, these findings suggest that the age-dependent decline of TET1, by altering the axo-myelinic junction, may further slowdown axonal conductance. Thus, despite the morphological evidence of remyelinated axons with the expected g-ratio, the newly generated myelin in older mice may not be functionally able to support effective conduction of action potentials, due to ionic and fluid imbalances at the axoglial junction, a concept which deserves future investigation.

Besides genes regulating ion homeostasis at the axo-glial junction, TET1-dependent hydroxymethylation regulates the expression of genes involved in metabolic processes, but not myelin genes. Accordingly, young adult *Tet1* mutants display normal myelin content and structure but myelin repair after a demyelinating injury is defective, possibly due to a combination of metabolic alterations and swellings of the axon–myelin interface. By regulating the levels of solute carriers and metabolic genes, DNA hydroxymethylation, mediated by TET1 in young adults guarantees a functional axon–myelinic interface and successful repair. As animals age, however, the declining levels of TET1 result in inefficient DNA hydroxymethylation during the phase of repair after demyelination. The decreased levels of TET1 gene targets, involved in the regulation of metabolites and ionic

exchanges create a dysfunctional axon–myelin interface, and further create functional impairments of the repair process.

Our study defines the major role of TET1-mediated DNA hydroxymethylation, as a regulator of gene expression during myelin repair, which becomes defective with aging. Worth noting, in this respect is the dramatic downregulation of the TET enzymes detected in post-mortem brain tissue from multiple sclerosis patients, thereby highlighting the overall translational potential of our findings[78].

## Methods

**Mouse models.** All experiments were performed according to approved protocols by the Institutional Animal Care and Use Committee (IACUC) of the Advanced Science Research Center (ASRC), City University of New York (CUNY). Mice were maintained in a temperature (65–75 °F) and humidity (40–60%) controlled facility on a 12-h light–dark cycle with food and water ad libitum. Mice from either sex were used and mutants were checked for survival and weight every day from birth to weaning. We used distinct reporter mouse lines for flow-activated cell-sorting: *Pdgfra-H2BEGFP* to sort oligodendrocyte progenitor cells[44] (RRID: IMSR_JAX:007669) and a previously characterized *Plp-EGFP* mouse to sort oligodendrocytes[43,45]. For conditional knock-out mouse lines, we crossed and bred the oligodendroglial-specific *Olig1-cre*[39] (RRID:IMSR_JAX:011105) line with the *Tet1-flox* (gift from. Pr. Yong-Hui Jiang)[40] or with the *Tet2-flox*[41] (RRID: IMSR_JAX:017573) lines. The *Tet1-flox* line was generated by flanking exon 4 with loxP sites[40], resulting in loss of exon 4 in OLIG1 + cells, and out-of-frame fusion of exons 3 and 5, yielding an unstable truncated product lacking the catalytic domain of TET1[79]. For conditional inducible deletion, we crossed and bred the oligodendroglial-specific *Pdgfra-creER(T)* [42] (RRID:IMSR_JAX:018280) line with the *Tet1-flox* line.

**Zebrafish husbandry and transgenic lines.** Adult zebrafish were housed and maintained in accordance with standard procedures in the Queen's Medical Research Institute zebrafish facility, University of Edinburgh. All experiments were performed in compliance with the UK Home Office, according to its regulations under project licenses 60/8436, 70/8436, and PP5258250. Adult zebrafish were subject to a 14/10 hours, light/dark cycle. Embryos were produced by pairwise matings and raised at 28.5 °C in 10 mM HEPES-buffered E3 Embryo medium or conditioned aquarium water with methylene blue. Embryos were staged according to days post-fertilization (dpf). For live imaging, the Tg(mbp:EGFP-CAAX) transgenic line was used. The *ue58* allele was identified during an ENU-based forward genetic screen as discussed in Marshall-Phelps et al.[52].

**Lysolecithin injection.** Injections were carried out in the ventrolateral spinal cord white matter of 8-week-old animals of either sex, as previously described[80]. Briefly, anesthesia was induced and maintained by inhalational of a mixture of isoflurane/oxygen. The vertebral column was fixed with metal bars onto the stereotaxic apparatus. The spinal vertebrae were exposed, tissue overlying the intervertebral space was cleared, and the dura pierced. A pulled glass needle was advanced through the spine, at an angle of 70°, and 1 µL of 1% lysolecithin (Sigma L4129) was slowly injected into the ventrolateral white matter. Mice were sutured and kept in a warm chamber during recovery. Mice were perfused at 4 day-post-lesion (4dpl), 7dpl, 14dpl, or 21dpl. For RNA-sequencing, control and lesioned fresh spinal cord tissue samples were collected at 4dpl. After mechanical dissociation, they were incubated in enzymatic dissociation medium containing papain (30 µg/ml in DMEM-Glutamax) and supplemented with 0.24 µg/ml L-cystein and 40 µg/ml DNase I. Dissociated samples were layered over a pre-formed Percoll density gradient and centrifuged for 15 min, to discard blood vessels, and enrich for oligodendroglial cell populations. Cells were washed twice in PBS 1X (PBS 10X, Invitrogen), then the dry cell pellets were frozen at −80 °C, and used for RNA extraction.

**Tamoxifen injections.** 4-Hydroxytamoxifen (Sigma-Aldrich, T56-48) was dissolved at 40 mg/ml in 10% ethanol and 90% corn oil (Sigma-Aldrich, C8267) for 4 h at 37 °C with rotation, and 10 mg were administered by gavage to each mouse at days 1, 3, and 5 before lysolecithin injection. Mice were perfused at 14dpl (day 25 from the first tamoxifen administration).

**Cell sorting.** Oligodendrocyte progenitor cells were isolated from post-natal day 5 (P5), P60 and P540 from *Pdgfra-H2BEGFP* brains[44], oligodendrocytes were sorted from P60 *Plp-EGFP* brains[44], using fluorescence-activated cell sorting as described previously[38]. Tissue was dissected in HBSS 1X (HBSS 10X (Invitrogen), 0.01 M HEPES buffer, 0.75% sodium bicarbonate (Invitrogen), and 1% penicillin/streptomycin) and mechanically dissociated. After an enzymatic dissociation step using papain (30 µg/ml in DMEM-Glutamax, with 0.24 µg/ml L-cysteine and 40 µg/ml DNase I), samples were layered over a pre-formed Percoll density gradient and then centrifuged for 15 min. Cells were then collected and stained with propidium iodide (PI) for 2 min at room temperature (RT). In a second step, GFP-positive and PI-negative cells were sorted by fluorescence-activated cell sorting (FACS; Aria, Becton Dickinson) and collected in pure fetal bovine serum. For RNA/DNA extraction, cells were washed twice in PBS 1X (PBS 10X, Invitrogen), then the dry cell pellets were frozen at −80 °C. For immunocytochemistry, cells were washed once in PBS 1X (PBS 10X, Invitrogen), then plated and maintained in Bottenstein and Sato medium (without growth or differentiation factors)[38] for 12 h before fixation.

**Western blot.** Mice of either sex were sacrificed at P14 or P60 and spinal cords were immediately processed for protein extraction. Protein lysates (50 µg) were separated by sodium dodecyl sulfate–polyacrylamide gel electrophoresis (SDS–PAGE) and transferred onto a PolyVinylidene DiFluoride (PVDF) (Millipore, Billerica, MA, USA) membrane using a buffer containing 25 mM Tris base, pH 8.3, 192 mM glycine, 20% (vol/vol) methanol for 1 h at 100 V at 4 °C. Membranes were blocked for 1 h in 10% Milk/0.1% Tween/TBS, then incubated overnight at 4 °C with the primary antibody diluted 1:1000 in 5% BSA/ 0.02% sodium azide/0.1% Tween/TBS. Antibodies used are: mouse anti-CNPase (Sternberger Inc., SMI-91, 1:5,000), mouse anti-MBP (Millipore, SMI-99, 1:5,000), mouse anti-GAPDH (Abcam, ab8245, 1:5,000). After rinsing with 0.1% Tween/TBS, membranes were incubated 2 h at room temperature with the secondary light-chain specific antibody (goat anti-mouse IgG, Jackson Immunoresearch, 115-035-174, 1:10,000) in 10% Milk/0.1% Tween/TBS. After rinsing, membranes were incubated with ECL (Amersham) for 3 min and then revealed. Quantification was carried out on three biological replicates per genotype, using ImageJ. Protein expressions were normalized to GAPDH expression, then compared to their respective expression levels in control samples. Uncropped gels are shown in Supplementary Fig. 9.

**Single-cell C1 capture and biomark.** For each age, sorted single-cells were directly captured on a freshly primed C1 integrated fluidic circuit (IFC) plate (10–17 µm diameter cells). Each plate was examined under a fluorescent microscope to identify single-cell and GFP + slots. RNA and cDNA were prepared using the C1 PreAmp protocol. P5, P60, and P540 single-cell and bulk samples were then harvested and combined on two 96 × 96 dynamic arrays IFC for C1 Biomark. The gene expression of 93 transcripts and 3 controls were analyzed on each of the 188 single-cell samples at the ages analyzed (P5: $n = 61$, P60: $n = 76$, P540: $n = 51$), 3 bulk samples and 1 negative bulk sample. C1 R package was used to analyze the normalized data and evaluate log2 Expression, derived from each Ct, using an ANOVA statistical test.

The primers used for the Biomark study are listed in Supplementary Table 2.

**DNA and RNA extraction.** DNA and RNA from FAC-sorted cells were isolated simultaneously from the same cell pellet using an AllPrep DNA/RNA Micro Kit (Qiagen) with on-column DNase treatment during the RNA isolation. Cortical and spinal cord tissues RNA were isolated from three biological replicates for each age and condition using TRIzol (Invitrogen) extraction and isopropanol precipitation. RNA samples were resuspended in water and further purified with RNeasy columns with on-column DNase treatment (Qiagen). RNA purity was assessed by measuring the A260/A280 ratio using a NanoDrop, and RNA quality checked using an Agilent 2100 Bioanalyzer (Agilent Technologies). DNA quality was checked using a NanoDrop and aQubit Fluorometric quantitation (Thermo Fisher).

**Quantitative real-time PCR.** For qRT-PCR, RNA was reverse-transcribed with qScript cDNA Supermix (Quanta) and performed using PerfeCTa SYBR Green FastMix, ROX (Quanta), at the ASRC Epigenomic Core. After normalization to the geometric mean of house-keeping genes, the average values for each transcript were calculated as based on the values obtained in all the samples included for each condition. A two-tailed Student's test was performed to assess statistical differences between the average values in each group. For overexpression assay, analysis was performed pair-wise for each cell culture replicate. The primers used are listed in Supplementary Table 3.

**RNA-sequencing.** For sorted cells, approximately 85 ng of total RNA per sample was used for library construction with the Ultra-Low-RNA-seq RNA Sample Prep Kit (Illumina) and sequenced using the Illumina HiSeq 2500 instrument according to the manufacturer's instructions for 100 bp paired end read runs. For oligodendroglial-enriched preparations from unlesioned and lesioned tissue, approximately 10 ng of total RNA per sample was used for library construction, using with the Ultra-Low-RNA-seq RNA Sample Prep Kit (Illumina) and then sequenced by the Illumina HiSeq 4000 instrument according to the manufacturer's instructions for 100 bp paired end read runs. High-quality reads were aligned to the mouse reference genome (mm10), RefSeq exons, splicing junctions, and contamination databases (ribosome and mitochondria sequences) using the Burrows-Wheeler Aligner (BWA) algorithm. Raw counts were generated with the Ensembl GRCm38 release 100, using uniquely aligned reads to exons by the featureCounts tool (subread 2.0.0). The raw read counts were input into DESeq2 v1.28.1[81], for normalizing the signal for each transcript and to ascertain differential gene expression with associated $p$-values. We used a cut-off of $p$-value $< 0.01$ and $q$-value $< 0.05$ to identified differentially expressed genes. For gene ontology analysis, we used GOrilla[82], Enrichr[83], and REVIGO[84]. We calculated the $p$-value

of the overlapped datasets using Fisher test in R (universe, total number of genes read on our datasets = 15,614). For the RNA-Sequencing from normal adult brains, the overall mapped reads and sample names are listed below. Data deposited in GEO: accession number GSE122446.

| Sample ID | Age | Tissue | Cell purification | Mapped read count |
|---|---|---|---|---|
| pdgfra2mn1 | P60 | Brain | *Pdgfrα-H2BEGFP* | 60,867,788 |
| pdgfra2mn2 | P60 | Brain | *Pdgfrα-H2BEGFP* | 57,999,332 |
| pdgfra2mn3 | P60 | Brain | *Pdgfrα-H2BEGFP* | 53,633,707 |
| plp2mn1 | P60 | Brain | *Plp-EGFP* | 51,987,110 |
| plp2mn2 | P60 | Brain | *Plp-EGFP* | 66,149,474 |
| plp2mn3 | P60 | Brain | *Plp-EGFP* | 62,479,413 |

For the RNA-sequencing data from the spinal cord of wild-type and *Tet1* mutants either unlesioned or after LPC injection, the mapped read counts are listed below. Data deposited in GEO: accession number GSE137611.

| Sample ID | Age | Tissue | Cell purification | Mapped read count |
|---|---|---|---|---|
| Tet1Control_SC_1 | P60 | *Olig1^{+/+};Tet1^{fl/fl}* control spinal cord | Percoll gradient | 77,725,400 |
| Tet1Control_SC_2 | P60 | *Olig1^{+/+};Tet1^{fl/fl}* control spinal cord | Percoll gradient | 62,825,397 |
| Tet1Control_SC_3 | P60 | *Olig1^{+/+};Tet1^{fl/fl}* control spinal cord | Percoll gradient | 69,992,533 |
| Tet1Control_LPC_2 | P60 | *Olig1^{+/+};Tet1^{fl/fl}* 4dpl spinal cord | Percoll gradient | 80,712,352 |
| Tet1Control_LPC_3 | P60 | *Olig1^{+/+};Tet1^{fl/fl}* 4dpl spinal cord | Percoll gradient | 82,184,360 |
| Tet1Control_LPC_4 | P60 | *Olig1^{+/+};Tet1^{fl/fl}* 4dpl spinal cord | Percoll gradient | 60,822,630 |
| Tet1cKO_SC_1 | P60 | *Olig1^{cre/+};Tet1^{fl/fl}* control spinal cord | Percoll gradient | 66,951,326 |
| Tet1cKO_SC_3 | P60 | *Olig1^{cre/+};Tet1^{fl/fl}* control spinal cord | Percoll gradient | 71,967,927 |
| Tet1cKO_SC_4 | P60 | *Olig1^{cre/+};Tet1^{fl/fl}* control spinal cord | Percoll gradient | 61,103,989 |
| Tet1cKO_LPC_1 | P60 | *Olig1^{cre/+};Tet1^{fl/fl}* 4dpl spinal cord | Percoll gradient | 60,348,900 |
| Tet1cKO_LPC_2 | P60 | *Olig1^{cre/+};Tet1^{fl/fl}* 4dpl spinal cord | Percoll gradient | 56,857,377 |
| Tet1cKO_LPC_3 | P60 | *Olig1^{cre/+};Tet1^{fl/fl}* 4dpl spinal cord | Percoll gradient | 78,129,031 |

**Reduced Representation Hydroxymethylation Profiling (RRHP)**. RRHP libraries were prepared from 200 ng input DNA per biological replicate, according to the manufacturer protocol (ZymoResearch, D5440)[85,86]. The raw sequencing reads were aligned to the mouse reference genome (mm10). DNA hydroxymethylation levels were calculated as the relative ratio of cytosine reads for each individual CpG site, for each sample, compared to all the replicates. Determination of differential hydroxymethylation was performed at the single base level, using only CpG dinucleotides. Differentially hydroxymethylated CpGs were selected at a $q$-value < 0.05. Differentially hydroxymethylated regions (DMR) were defined by 2 kb genomic regions containing at least 2 concordantly hydroxymethylated CpGs (in the same direction). Genes with differentially hydroxymethylated CpGs ($q$-value < 0.05) were compared to differentially expressed genes ($p$-value < 0.01 and $q$-value < 0.05) to generate a scatterplot using smoothScatter in R. For the gene region analysis, all the sites that fell within a gene body were linked to that given gene, and a table of pairs of RRHP sites and genes was produced. The same was done for the promoter analysis, but instead of gene bodies, a flanking region of 2 kb upstream from the gene's transcription start site to 500 bp downstream from the TSS was used as the region for pairing. All the RRHP data from the above samples have been deposited in GEO: accession number GSE122446.

| Sample ID | Age | Tissue | Cell purification | RRHP read count | Unique CpGs |
|---|---|---|---|---|---|
| pdgfra2mn1 | P60 | Brain | *Pdgfrα-EGFP* | 13,886,881 | 980,401 |
| pdgfra2mn2 | P60 | Brain | *Pdgfrα-EGFP* | 16,214,208 | 1,005,400 |
| pdgfra2mn3 | P60 | Brain | *Pdgfrα-EGFP* | 24,034,708 | 1,111,395 |
| plp2mn1 | P60 | Brain | *Plp-EGFP* | 37,707,720 | 1,295,473 |
| plp2mn2 | P60 | Brain | *Plp-EGFP* | 21,883,753 | 1,103,851 |
| plp2mn3 | P60 | Brain | *Plp-EGFP* | 35,946,466 | 1,270,551 |

**Mixed neuron-glia myelinating cultures**. Forebrains were removed from 15-day mouse fetuses, dissociated first mechanically and then enzymatically with 0.05% Trypsin in HBSS 1X for 15mn at 37 °C. Cells were washed and filtered gently through nylon mesh (63 μm pores). The pellet was suspended in DMEM/10% FBS. Cells were plated on poly-L-Lysine coated glass coverslip. The culture medium consisted of Bottenstein and Sato medium supplemented with 1% FBS, 1%penicillin/streptomycin solution, PDGF-AA (10 ng/ml), with addition of T3 hormone (45 nM) after the third day in culture, to induce differentiation.

**Primary oligodendrocyte progenitor cultures and overexpression studies**. Mouse OPCs were isolated from the brain of P7 mice through immunopanning with a rat anti-mouse CD140a antibody (BD Biosciences, 558774), recognizing PDGFRα[87]. Cells were plated in 10-cm dishes and kept proliferating in PDGF-AA (10 ng/ml) and bFGF (20 ng/ml). The catalytic domain of human TET1 (hTET1-CD) was amplified from a template plasmid, pIRES-hrGFP II-TET1_CD (RRID: Addgene_853570)[88], and then sub-cloned into the linearized lentiviral vector, pScalps_Puro (RRID:Addgene_99636)[89], by using In-Fusion cloning system (Takara Bio). The empty lenviral backbone was used as negative control (CTRL). OPCs were transfected with 10 μL of CTRL or hTET1-CD lentivirus overnight. After 24 h, media was changed and cells were maintained in proliferative media for 5 days before fixation for immunocytochemistry or collection for qRT-PCR.

**Immunocytochemistry**. After fixation, cells were incubated in blocking buffer (10% normal goat serum in PBS/Triton 0.3%) for 1 h at room temperature and then incubated overnight at 4 °C with the primary antibodies diluted in the same blocking buffer. Cells were incubated with the appropriate Alexa Fluor conjugated secondary antibodies for 1 h at room temperature, then mounted using Fluoromount-G with DAPI.

*Primary antibodies used for immunocytochemistry.* Primary antibodies used for immunocytochemistry were rat anti-NFM-H (Millipore, MAB5448, 1:500), rabbit anti-NG2 (Millipore, AB5320, 1:200), mouse anti-SLC12A2 (DSHB, T4, 1:100), rat anti-PDGFRα (Millipore, CBL1366, 1:800), rat anti-MBP (Abd Serotec, MCA095, 1:200), mouse anti-MOG (Millipore, MAB5680, 1:500), mouse anti-OLIG2 (Millipore, MABN50, 1:200), rabbit anti-TET1 (Novus Biologicals, NBP1-78966, 1:100).

*Secondary antibodies used for immunocytochemistry.* Secondary antibodies used for immunocytochemistry were goat anti-rat (IgG 647, Life Technologies, A21247, 1:1,000), goat anti-mouse (IgG 546, Invitrogen, A11030, 1:,1000), goat anti-rabbit (IgG 488, Invitrogen, A11034, 1:1,000), goat anti-rabbit (IgG 546, Invitrogen, A11035, 1:1,000), goat anti-mouse (IgG 488, Invitrogen, A11029, 1:1,000), goat anti-rat (IgG 555, Invitrogen, A21434, 1:1,000).

**Immunohistochemisty**. For immunohistochemistry, animals were perfused with 4% paraformaldehyde and post-fixed overnight in the same solution at 4 °C. Brains and spinal cords were dissected, cryo-protected in sucrose solutions, and frozen embedded in OCT. Immunohistochemistry was performed on 12 μm cryostat sections. For 5mC and 5hmC staining, slides were first permeabilized, denaturalized, and neutralized. Slides were incubated in blocking buffer (5% normal goat serum in PBS/Triton-X100 0.3%) for 1 h at room temperature and then overnight at 4 °C with the primary antibodies diluted in a similar blocking buffer (5% normal goat serum in PBS/Triton-X100 0.3%). After rinsing with PBS 1X, sections were incubated with the Alexa Fluor secondary antibodies and then washed with PBS 1X. Stained tissue were cover-slipped in DAPI Fluoromount G mounting medium (Thermo Fisher) and examined on a Zeiss LSM800 Fluorescence Microscope. For 5mC and 5hmC quantifications, the 5mC and 5hmC immunoreactivity of each antibody per nuclear area (pixel/μm$^2$) was quantified for at least 50 OLIG2 + cells in each technical replicate. We used three technical replicates per biological conditions/ages. Cells were qualified as low (<5 pixel/μm$^2$), medium (5–10 pixel/μm$^2$) or high (>10 pixel/μm$^2$) immunoreactivity relative to OLIG2 nuclear intensity.

*Primary antibodies used for immunohistochemistry.* Primary antibodies used for immunohistochemistry were Fluoromyelin Green Fluorescent Myelin Stain (Invitrogen, F34651, 1:300), mouse anti-5mC (Abcam, ab10805, 1:200), rabbit anti-5hmC (Active Motif, 39769, 1:200), mouse anti-CC1 (Millipore, OP80, 1:200), rat anti-MBP (Abd Serotec, MCA095, 1:200), mouse anti-OLIG2 (Millipore, MABN50, 1:500), rabbit anti-OLIG2 (Santa Cruz, sc48817, 1:200), mouse anti-SLC12A2 (DSHB,T4, 1:100), rabbit anti-TET1 (Novus Biologicals, NBP1-78966, 1:100), rabbit anti-TET2 (Epigentek, A-1701, 1:100), rabbit anti-TET3 (Abcam, ab139311, 1:100), rabbit anti-KI67 (Abcam, ab21700, prediluted), mouse anti-GFAP (BioLegend, 644701, 1:200), mouse anti-ISLET1 (Abcam, ab86472, 1:200).

*Secondary antibodies used for immunohistochemistry.* Secondary antibodies used for immunohistochemistry were goat anti-mouse (IgG 488, Invitrogen, A11029, 1:1,000), goat anti-rabbit (IgG 488, Invitrogen, A11034, 1:1,000), goat anti-rabbit (IgG 546, Invitrogen, A11035, 1:1,000), goat anti-mouse (IgG 546, Invitrogen, A11030, 1:,1000), goat anti-rat (IgG 647, Life Technologies, A21247, 1:1,000).

**Electron microscopy**. For electron microscopy, animals were perfused at P14 or at P60 at 14dpl or 21dpl with 4% glutaraldehyde in PBS containing 0.4 mM CaCl2 and post-fixed in the same solution at 4 °C. The spinal cord was coronally sliced at 1 mm thickness and treated with 2% osmium tetroxide overnight before being subjected to a standard protocol for epoxy resin embedding. Tissues were sectioned at 1 μm and stained with toluidine blue. Ultrathin sections of the lesion site were cut onto copper grids and stained with uranyl acetate before being examined with a Hitachi H-600 transmission electron microscope. Quantification was performed on 50 nm sections on a minimum of 80 axons per animal, three or four mice for each genotype. G-ratio curves were fit with non-linear regression between control and *Tet1* mutant replicates and slopes compared with sum-of-squares F Test.

**Immunoelectron microscopy**. For immunoelectron microscopy, animals were perfused with 4% paraformaldehyde and 0.2% glutaraldehyde in 0.1 M phosphate buffer containing 0.5% NaCl, as previously described[90]. For immunolabeling, optic nerve sections were incubated with mouse anti-SLC12A2 (DSHB, T4, 1:2,000) in blocking buffer, followed by five washes with PBS and 20 min incubation with protein A-gold (10 nm) in blocking buffer.

**Zebrafish live imaging and single cell labeling**. Live imaging of the Tg(mbp: EGFP-CAAX) line was carried out using a Zeiss 880 LSM with Airyscan in super-resolution mode, using a 20X objective lens (NA = 0.8). Larvae were embedded in 1.5% low melting point agarose in embryo medium with tricaine. To mosaically label individual oligodendrocytes in slc12a2b[ue58] mutants and siblings, we injected one-cell staged embryos with 1 nl of a solution containing 10 ng/μl pTol2-mbp: memScarlet plasmid DNA and 25 ng/μl tol2 transposase mRNA.

**Quantification and statistical analysis**. Image acquisition was performed on a Zeiss LSM800 Fluorescence Microscope, using ZEN software. Images were analyzed with Fiji-Image J (RRID:SCR_003070). All statistical analyses were done using GraphPad Prism (GraphPad Software, Inc, RRID:SCR_002798). Unpaired Student's *t* test two-tailed was used for every two datasets with equal variances and for which data follow a normal distribution. One-way and two-way ANOVA was used to compare three or more sets of data. For all graphs, error bars are mean ± SEM. For all quantifications, $n = 3–7$ mice were examined (3–6 images were analyzed and averaged per mouse, for each staining). All statistical details for each graph can be found in the figure legends. qRT-PCR on tissues and single-cells have been replicated twice in independent experiments. All immunostaining on P14 *Tet1* mutants and quantifications have been duplicated and evaluated by two independent investigators[45,91–93].

**Reporting summary**. Further information on research design is available in the Nature Research Reporting Summary linked to this article.

## Data availability

All data supporting the findings of this study are provided within the paper and its supplementary information. A source data file is provided with this paper. RNA sequencing and Reduced Representation Hydroxymethylation Profiling data from wild-type adult mice have been deposited in NCBI's Gene Expression Omnibus Series accession number GSE122446. RNA sequencing from wild-type and *Tet1* mutants during myelin repair accession number GSE137611. Previously published and deposited data used in this study: GSE48872. Lineage specific genes were exported for these microarray data and organized using hierarchical clustering on Multi-experiment Viewer. Further information and requests for resources and reagents should be directed to and will be fulfilled by Dr. Patrizia Casaccia. N.A. Source data are provided with this paper.

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

## Acknowledgements

This work was supported by R35 NS111604-01 and NMSS-RG4890A10 to PC, post-doctoral fellowships from the Paralyzed Veterans of America (3061) and from National Multiple Sclerosis Society (FG-1507-04996) to SM. BS was funded through

the Cluster of Excellence and DFG Research Center Nanoscale Microscopy and Molecular Physiology of the Brain. Work in the Lyons lab was supported by Wellcome Trust Senior Research Fellowships (102836/Z/13/Z and 214244/Z/18/Z) and by a project grant from the Multiple Sclerosis Society (95). We thank the Flow Cytometry Core CyPS (Pierre & Marie Curie University, Pitié-Salpêtrière Hospital, Paris, France) for help on sorting adult OPC and adult OL used for sequencing, the Epigenomic Core at the ASRC-CUNY for OPC sorting and C1/Biomark experiments, the Epigenomics Core at Weill Cornell for sequencings and the NYU EM Core for electron microscopy on LPC tissues. We would like to thank the Casaccia lab members for fruitful discussions.

## Author contributions

Data acquisition, S.M., R.F., K.M.P., L.K., S.B., D.M., D.H., B.S., D.A.L., W.M.; Experimental design, validation, data analysis, visualization, original draft, S.M.; Conceptualization, writing, revisions, and editing, S.M. and P.C.; Funding acquisition, supervision, P.C.; Resources, Y.H.J.

## Competing interests

The authors declare no competing interests.
