## [Peer Review File · Nature Communications]

Reviewers' Comments:

Reviewer #1:

Remarks to the Author:

In the current manuscript the authors identify TET1 and DNA hydroxy methylation as necessary for adult OPC differentiation into OLs during myelin regeneration in adult mice. This follows their previous and similar analyses of DNMT1 being necessary for DNA methylation and OPC differentiation in mice during development (cell reports) and DNMT3 being necessary for DNA methylation and OPC differentiation during myelin regeneration in adult mice (eneuro). As the authors state it is well known that the global epigenome changes with aging and in addition it is well known that global disruption of the epigenome interferes with differentiation which has now been shown in many different developmental systems for a host of epigenetic regulators. In addition to DNA methylation, OPCs also fail to differentiate when other epigenomic marks such as acetylation or methylation of histones or key transcription factors are perturbed. It is therefore not at all surprising that interfering with global DNA hydroxy methylation has a similar albeit more subtle effect. Thus, without a rescue experiment in aged mice (TET1 overexpression), the link between hydroxy methylation SLC12a2 and aging remain correlative and expected, indicative of sick cells that fail to execute differentiation programs. While doing a rescue experiment may be too much to ask in the current global shutdown, the role between global epigenomic stress and differentiation should be fairly confronted.

The authors also use an elaborate expression analysis to justify their interest in hydroxy methylation. These analyses are not very well controlled, and many details are missing which is a shame as a venture into hydroxy methylation could also be simply justified from their previous work on 5mC methylation.

For instance:

- The authors compare adult mice (day 60 OPCs vs day 60 OLs) with post-natal mice from different ages (day 5 OPCs and day 16 OLs). They note differences in the number of genes that are differentially expressed and differences in their GO annotations. How much of this is due to batch/lab, age differences in the neonatal mice or actual differences between OPCs and OLs is unclear. Also why is there an imbalance in up/down for the P60 data (2k vs 1k) and not the published data (4k vs 4k). Are there quality differences? The use of a combined score (p value and zscore) for GO analysis is something I haven't seen before. Multiple testing corrected p values should do here.

- The authors then compare genes that are different between aOPCs and nOPCs with epigenetic marks that are different between nOPCs and nOLs. This is comparing apples and oranges. Why is an nOL a good proxy for aOPCs?

- The authors then show in this analysis that several of the genes that change between aOPCs and nOPCs overlap several epigenetic marks that also change, and most are DNA methylation. However, there is no analysis of this number in relationship to the total number of marks and thus the reader has no idea what these numbers represent. Are any of these overlaps significant or can this all be explained by random chance? Also, what does overlap mean, does the promoter need to be overlapped or a gene body, or is the analysis vicinity based.

- The authors continue to compare 5mC with hydroxy methylation using immunofluorescence and state that hydroxy methylation is more affected than 5mC. While for my part the manuscript could have started here, or even after this analysis, can we really quantify two separate antibodies using arbitrary cutoffs like this?

The authors then move into their mouse experiments using several crosses and the experiments that show a role for TET1 and not TET2 in adult myelination. These experiments are much better controlled and show that global reduction of hydroxymethylation interferes with differentiation during remyelination.

Then the authors use reduced representation hydroxy methylation profiling and RNA Seq to analyze their system further. There's a difference in the number of transcripts that are up and downregulated pre and post lesion in TET mutant cells (less differences) compared to controls.

Again, several numbers are thrown at the reader lacking context and statistics. For instance

5583 genes are associated with hydroxy DMRs of 2kb with at least two cpGs in the gene region. How are DMRs coupled to genes, how is the gene region defined. Do longer genes that are typically overrepresented in neuronal genes hold more DMRs and is this controlled for? Is a single base shift sufficient to call a region DMR? (two for the 5583 genes ?).

The overlap between increased hydroxy methylated regions and upregulated transcripts is 323. Is this significant? what does this number tell us? A significant correlation would be expected between altered hydroxy methylation and altered expression. Is altered hydroxy methylation linked to the promoter regions of these genes ?

Slc genes are overrepresented, but these are large genes (Slc12a2 is ~200 kbp). Is this corrected for?

Reviewer #2:

Remarks to the Author:

Moyon et al. characterized oligodendrocyte (OL) lineage-specific conditional KO mice of TET1, the primary enzyme responsible for DNA hydroxymethylation in the CNS, and defined its role in remyelination. Combined with analyses of diverse OL and OPC-specific RNAseq datasets, the authors suggest the link between reduced TET1 levels and reduced remyelination efficiency that are commonly found in the aged CNS. The authors compared the RNAseq results from control and Tet1 KO mice, and proposed some TET1 target genes including SLC12A2, and showed that a loss of function mutant of slc12a2b, a homolog in zebrafish led to oligodendroglial defects. With these results, Moyon et al. concluded that TET1 regulates oligodendrocyte differentiation and remyelination.

The DNA hydroxymethylation mediated by the TET family may be an important epigenetic mechanism for adult oligodendroglial regulation, and this study seems to be the first to assess the contribution of this class of genes to normal myelination and adult myelin repair. However, the phenotypes of the mutant mice were somewhat weak and may require additional data to support the authors' conclusion that Tet1 critically regulates adult remyelination after demyelinating injury, and may provide the more detailed cellular mechanism.

I would recommend some revisions to address some questions and ultimately improve the paper.

1. There are concerns about mice used for OPC and OL isolation and subsequent RNAseq, and Olig1-Cre mice because of their transgene expression (or activity) outside target cells or lineages. Therefore, additional supplemental figures should be prepared to show their target cell specificity at the examined ages. Relevant fractional quantification should be performed, and results interpretation should be cautiously made.

a. The PDGFRa-EGFP (knock-in mice with a sacrifice of one allele of Pdgfra) is known to express EGFP not only in OPCs but also in PDGFRa+ pericytes (Kang et al., 2010) or leptomeningeal cells (Marques et al. 2016; Marques et al., 2018). Therefore, the fraction of EGFP+ cells that express other than NG2+ or Olig2+ cells at the studied ages (P6, P60, and P540).

b. The Plp-EGFP Tg mice used in this study, (if these were generated by Wendy Macklin group) are known to express EGFP in NG2+ OPCs in addition to oligodendrocytes (OLs) (Mallon et al., 2002). If this line of mice were used, EGFP expression in OLs and OPCs should be carefully quantified for the target ages (P16, P60, P540) in the target brain areas. Also, the ref. (#53) does not appear to be the correct source for Plp-EGFP mice. #53 (Spassky et al., 1998) reported the generation of Plp-LacZ Tg mice.

c. Olig1-Cre (knock-in with a sacrifice of one allele of Olig1) was reported to target most astrocytes in addition to oligodendrocyte lineage (De Biase et al., 2011). Images of cre reporter

(e.g., Ai14) expression after crossing Olig1-Cre should be shown, and if the Cre activity is broadly observed other than OL lineage, the results should be stated accordingly.

2. The authors show that oligodendroglial Tet1 (Olig1-Cre; Tet1 f/f) ablation did not impact developmental myelination (Figure 4). Even when remyelination defects were observed in the same cKO (Figure 5), the numbers (or percentage) of CC1+ cells and Olig2+ cells did not decrease (Supplemental Figure 4), indicating that the aOPC > OL differentiation is not disturbed by the Tet1 ablation. Combined together, the inefficient recovery of myelin (based on weaker MBP immunoreactivity, fluoromyelin patterns, and reduced percentage of newly myelinated axons (14 dpl) shown in Figure 5) suggest that Tet1 is required for myelinogenesis (or myelin maintenance), but not for oligodendrogenesis during the regenerative process. However, the authors state "Our results identify DNA hydroxy-methylation as necessary for aOPC differentiation". Change the statements or provide supporting evidence.

3. In Figure 3, does the inefficient myelin repair observed in old mice involve impaired differentiation of OPC to OLs, or only inefficient myelination from newly formed OLs?

4. What is the cell stage in which TET1 exerts its effect? The authors assumed that the cell stage for Tet1 activity is OPC throughout the manuscript. But there is little or no evidence suggesting that TET1 impact on OPCs, or 'aOPC to OL differentiation'. As pointed above, only myelination defects were seen during remyelination in the Tet1 mutant mice. If Tet1 ablation impacts only myelination without impaired new OL generation, Tet1 may play roles in OLs for myelin (membrane) growth or myelin maintenance. The authors should clearly distinguish the two possible scenarios. Can Tet1 or 5hmC be co-immunostained with NG2 and CC1, not only with Olig2, during remyelination process?

5. In Figure 6, the degree of Slc12a2 expression reduction in Tet1 cKO mice is much greater in the spinal cord tissue (Figure 6E) than in FACS-isolated OPCs (Figure 6F). Does this indicate that Tet1 ablation more impacts OLs (or other cells) than in OPCs?

6. In Figures 7D and 8C, after LPC, does the upregulation of Slc12a2 occur only in oligodendroglia, or in astrocytes and microglia too? What are the dashed lines (Are they expressed in gray matter)? The upregulated Slc12a2 signals during remyelination should be examined with cell stage markers (NG2 and CC1) and cell type-specific markers (astrocytes and microglia).

7. In Figure 8, although zebrafish slc12a2b is a homolog of Slc12a2, it is a paralog of slc12a2a but not the ortholog of Slc12a2. How about the phenotype of loss of function mutant of slc12a2a? Does Tet1 also regulate hydroxymethylation of slc12a2b in zebrafish?

8. It is not clear whether the EM results of Tet1 mutation (Figure 8E) correspond to the abnormalities (shown in Figure 8I) out of slc12a2b mutation in zebrafish, as one came from EM and the other from low-power magnified fluorescence. It will be helpful to show the phenotypes observed in zebrafish is similar or identical to the mouse at EM levels. Moreover, the phenotype of the mutant in zebrafish appeared during normal early development. However, TET1 KO conditional mutant mice did not show any abnormalities in myelination. Explain this.

9. In Figure 8, the mbp promoter was used to express a fluorescence protein in zebrafish, which allowed for the label of both peripheral and central myelin. Thus, are those abnormalities shown in Figure 8G- I related to Schwann cells and OLs. In ref # 57, the co-authors reported this phenotype as peripheral myelin swelling. It would be great if the authors clearly state how this experiment results (Figure 8) are different from those of ref # 57. Are the myelin defects in zebrafish mutants caused by cell-autonomous or non-cell autonomous mechanisms?

Minor :

10. Grammar error (in Page 10. first paragraph) "Demyelination was induced in the spinal cord of inducible Tet1 mutants at P60, one week after tamoxifen induction by gavage (Figure 5I). and Control mice, lacking cre expression, were subject to the same experimental protocol." Remove 'and' between the two separate sentences.

11. What brain areas of Tg mice did the authors collect to isolate OPCs and OLs using FACS?
12. In Figure 6A, in the upper diagram, P60 control mice is also denoted as Olig1-Cre; Tet1 flox, which is the same as mutant mice. Correct this.
13. In Figure 7A, provide separate each colored images, as least for the magnified view, while the merged image remains. Consider that the NFM (neuro-filament M) label to be white font in black background or use a similar alternative.
14. In Figure 7D, are the number of Olig2+ cells smaller in old mice? Then the low level of Slc12a2 is not caused by the decreased expression, but by the decreased OPC recruitment?
15. In Figure 7E, there are no error bars in the white column (young age), while the bar has multiple values (dots)?
16. Page 12 – 13: The authors use terms somewhat inconsistently: neuroglia interface, axo-glial interface, axo-glial junction, axo-myelin junction, axon-myelin interface (page 16). 'Axo-glial junction' is a specialized term that describes cell adhesion flaking node of Ranvier and involves the interactions of multiple proteins. It is not clear if the provided EM data indicate disruption of axo-glial junction.
17. The authors stated: "We identified the gene ontology category "neuronal-oligodendroglial communication", which included solute carrier transporters. We focused on the solute carrier gene family, especially on the member Slc12a2." Provide GO term number for the major GOs in the text or figure legends.
18. (Page 16) "The detection of SLC12A2 at the axon-myelin interface was consistent with the concept that this is an electrically silent ionic transporter whose function is to regulate ionic balance and cell volume." If there is a reference that showed NKCC1 regulates OL (or myelin sheath) volume, please cite it.
19. A recent study (Jantzie et al., 2015) suggests that inhibition of NKCC1 benefits oligodendrocyte (or WM preservation or repair after injury) in a mouse model of PVL, and so appears to be conflicting to the current report. Consider discussing this paper.
20. The title of some figures and subheadings are far moved from objective data, so somewhat subjective or not consistent with data. In Supplemental Figure 3, the title is "Constitutive ablation of Tet1 in OPC mimics the events leading to defective myelin regeneration detected in old mice." However, the data did not show any abnormalities in the three assessed parameters (percentages of CC1+ cells, Olig2+ cells, and remyelinated axons) after Olig1-Cre-mediated Tet1 cKO. It is not clear about what events are related to defective myelination.

Reviewer #3:

Remarks to the Author:

The manuscript by Moyon et al identified a novel and necessary role of TET1-mediated DNA hydroxymethylation in adult myelin repair. The authors investigated the three TET family enzyme levels change with age, and they found DNA hydroxymethylation and increased expression of Tet1 were detected in young adult mice after demyelination, but not in the old mice. Specific ablation of Tet1 but not Tet2 led to the age-related decrease of DNA hydroxymethylation and inefficient remyelination. The authors also performed genome-wide hydroxymethylation and transcriptomic analysis to identify the Tet1 target genes including several members of the solute carrier (Slc) gene family. They found that transcript levels of Slc genes including Slc12a2 were lower in Tet1 mutants and old mice and were also associated with swelling at the neuroglial interface detected in zebrafish mutants. Overall, this work is well-executed, and the data is presented logically.

Main comments:

1. The authors showed experimentally that compared to Tet1, Tet2 does not play a major role in adult remyelination. However, it is questionable how did the authors rule out Tet3. The reasons the authors provided are 'Tet3 levels showing only minimal changes with age (Figure 3M)' and 'we were unable to detect TET3 in OPCs or OLs in the adult mouse brain, using multiple approaches. This finding is consistent with previous reports on TET3 expression limited to embryonic tissues.' These are just not true.

First, according to Figure 3M, Tet3 levels showed the largest changes with age in OPCs (smallest ANOVA p-value), even stronger than Tet1.

Second, again according to Figure 3M, Tet3 is clearly detectable in adult OPCs, showing higher expression than Tet1. Figure 2F, on the other hand, showed Tet3 is barely detectable in OLIG2+, seems to be contradictory to Figure 3M. However, 2F and 3M are different techniques and measurements. I suggest cross-validating 2F and 3M with the other method.

Third, on the contrary, many reports showing Tet3 is was the most expressed Tet enzyme in mouse brain (e.g. doi.org/10.1093/nar/gkq684), which agrees with the data showing in Figure 3M. Therefore, the authors need to explain convincingly why they eliminated Tet3 or to conduct the experiments for Tet3 as they did for Tet1 and Tet2.

2. It is very interesting that the depletion of Tet1 only affects adult myelination but not developmental myelination. The authors mentioned very briefly DNMT might be the reason. I think more discussion should be included to show why it is the case.

3. The authors used RRHP to profile the genome-wide hydroxymethylation but did not show any results from RRHP directly. RRHP is not the most commonly used technique to sequence 5hmC. It will be good to add a supplementary figure to show the general analysis from RRHP, for example, the 5hmC distribution is consistent with previous 5hmC sequencing in mouse brain obtained with other methods.

Minor comments:

1. There is no need to add the hyphen in 'methyl-cytosine' and 'hydroxy-methylation'.

2. Page 3, "These enzymes act as methyl-cytosine dioxygenases, which convert a hydrogen atom at the C5-position of cytosine to a hydroxy-methyl group (5hmC), through oxidation of 5mC" is chemically incorrect. It should be "...convert a hydrogen atom of the methyl group at the C5-position of cytosine to a hydroxyl group...".

3. Images in Figure 1 should have a higher resolution.

4. Page 7, "The percentage of OLIG2+ cells expressing TET1 (Supplemental Figure 1A-D) and TET2 (Supplemental Figure 1E-H) in young and old spinal...". TET1 (Supplemental Figure 1A-D) should be TET1 (Supplemental Figure 1A, B, E, and F); TET2 (Supplemental Figure 1E-H) should be TET2 (Supplemental Figure 1C, D, G, and H)

5. Figure 4J, the top and bottom panels have the same label (olig1+/+; Tet1fl/fl).

Rebuttal to Reviewers' comments:

We thank the Editor and the Reviewers for their comments and for understanding that the pandemic and additional social crisis in NYC have brutally crippled our research operation. We have appreciated the clear delineation of the strengths and weaknesses of the previous submission. It allowed us to address all the potential areas of weakness, while providing additional data to support the main message related to the essential role of TET1-dependent DNA hydroxymethylation for myelin repair and highlight the importance and novelty of the current study.

Reviewers 1 and 3 provided important feedback on the molecular/epigenomic component of the paper while Reviewer 2 focused on the cellular aspect. Based on their comments, we substantially reorganized the text and figures of the manuscript, to clarify the concept of DNA hydroxymethylation as an age-dependent epigenomic modification that is dispensable for developmental myelination, but is important for activating the expression of genes regulating adult oligodendrocyte progenitor cells (OPCs) during myelin repair after injury. More specifically, with the exception of Figures 7 and 8 which remain almost untouched, we revised or replaced most of the figures and added several new supplemental figures. In response to Reviewers 1 and 3 and Editor's comments we reorganized the entire analysis of the molecular datasets, as reflected in the new figures. In this amended version, we omit datasets obtained in cultured neonatal OPCs and include a deeper analysis of the DNA hydroxymethylation (Reduced Representation Hydroxymethylation Profiling) and transcriptomic (RNA-Sequencing) datasets obtained from cells isolated from adult reporter mice. In addition, we focus on the identification of gene targets regulated by the hydroxymethyltransferase TET1 during myelin repair. We analyze the overlap between the RNA transcriptomic datasets obtained from adult control and *Tet1* null oligodendroglial preparations from the spinal cord, either unlesioned or after a demyelinating lesion. In response to Reviewer 2 and Editor's comment we revised most of the main and supplemental figures and added two new supplemental figures to directly address the cellular characterization of the mouse mutants and to provide a detailed characterization of the sorted cell populations.

Due to the extensive revisions, we have omitted the track change feature, as the text has been thoroughly edited. Below is a point-to-point rebuttal (regular font size) to the Reviewers comments (*in bold and italics*). We sincerely hope that this extensive revision will satisfy all the previous Reviewers' and Editor's concerns.

Reviewer #1 (Remarks to the Author):

1. *In the current manuscript the authors identify TET1 and DNA hydroxy methylation as necessary for adult OPC differentiation into OLs during myelin regeneration in adult mice. This follows their previous and similar analyses of DNMT1 being necessary for of DNA methylation and OPC differentiation in mice during development (cell reports) and DNMT3 being necessary for DNA methylation and OPC differentiation during myelin regeneration in adult mice (eneuro).*

In agreement with the Reviewer's statement, we have reorganized the introduction to further highlight this concept.

2. *As the authors state it is well known that the global epigenome changes with aging and in addition it is well known that global disruption of the epigenome interferes with differentiation which has now been shown in many different developmental systems for a host of epigenetic regulators. In addition to DNA methylation, OPCs also fail to differentiate when other epigenomic marks such as acetylation or methylation of histones or key transcription factors are perturbed. It is therefore not at all surprising that interfering with global DNA hydroxy methylation has a similar albeit more subtle effect. Thus, without a rescue experiment in aged mice (TET1 overexpression), the link between hydroxy methylation SLC12a2 and aging remain correlative and expected, indicative of sick cells that fail to execute differentiation programs.*

We are grateful to this Reviewer for bringing to our attention a very important concept that was clearly not highlighted or explained in the original submission. The idea that epigenomic changes affect oligodendrocyte differentiation, is indeed well-accepted. However, epigenetics encompasses a large number of processes, including histone modifications, non-coding RNAs, micro-RNAs, chromatin remodeling and DNA modifications, which differentially impact the transcriptome of oligodendroglial lineage cells. Most of previous studies addressed the importance of epigenomic changes consistent with transcriptional repression, leading to a model of nOPC differentiation driven by the repression of negative regulator during developmental differentiation. Here, in contrast, we show that TET1-dependent DNA hydroxymethylation drives the transcriptional activation of several genes involved in myelin repair, including genes regulating lipid metabolism and cell-cell communication (new Figures 5 and 6). This concept is now clearly discussed in the result section (pages 8-11) as well as in the

discussion. Worth noting is the fact that hydroxymethylation does not contribute to myelin gene transcriptional activation (please see also Table 1). In addition, the similar number of OLIG2 and CC1 positive cells in *Tet1* mutants and controls argues against the concept of “*sick cells that fail to execute differentiation program*” and suggests that hydroxymethylation is responsible for activating distinct gene ontology categories affecting the repair process after demyelination. More specifically it appears to coordinate the expression of enzymes involved in lipid metabolism and genes involved in cell-cell communication, including solute carriers.

- 3. While doing a rescue experiment may be too much to ask in the current global shutdown, the role between global epigenomic stress and differentiation should be fairly confronted. The authors also use an elaborate expression analysis to justify their interest in hydroxy methylation. These analyses are not very well controlled, and many details are missing which is a shame as a venture into hydroxy methylation could also be simply justified from their previous work on 5mC methylation. For instance: The authors compare adult mice (day 60 OPCs vs day 60 OLs) with post-natal mice from different ages (day 5 OPCs and day 16 OLs). They note differences in the number of genes that are differentially expressed and differences in their GO annotations. How much of this is due to batch/lab, age differences in the neonatal mice or actual differences between OPCs and OLs is unclear. Also why is there an imbalance in up/down for the P60 data (2k vs 1k) and not the published data (4k vs 4k). Are there quality differences? The use of a combined score (p value and zscore) for GO analysis is something I haven't seen before. Multiple testing corrected p values should do here. The authors then compare genes that are different between aOPCs and nOPCs with epigenetic marks that are different between nOPCs and nOLs. This is comparing apples and oranges. Why is an nOL a good proxy for aOPCs?***

This is an excellent point, which has been thoroughly taken in consideration. Since the epigenome results from the response of cells to environmental cues, it is reasonable only to compare epigenetic marks detected in specific cell types within a similar environment. As such, since our study is entirely conducted *in vivo*, we have excluded any comparison with previously published datasets obtained in cultured neonatal cells and replaced the previous Figures 1 and 2. We now entirely focus our manuscript on adult oligodendroglial lineage cells. However, we would like to highlight that neonatal and adult OPC samples were all sequenced at the same time, using a similar pipeline. Those neonatal and adult samples were processed for RNA-Sequencing, which yielded similar read counts (averaging 50,000,000 reads/sample) and they were similarly processed to identify differentially expressed genes using the same cut-offs (i.e. p value <0.01 and q value <0.05). Therefore, even though in the current manuscript we omit those data, we believe that the differences between developmental and adult oligodendrocyte differentiation presented and discussed in the first submission, reflected biological differences, rather than technical artifacts. In this revised manuscript, we have focused the adult oligodendrocyte differentiation dataset at the gene level using DESeq2 v1.28.1 and identified 1,187 down-regulated and 1,739 up-regulated genes in adult OLs compared to adult OPCs. Those data are now included in the new Figure 5.

- 4. The authors then show in this analysis that several of the genes that change between aOPCs and nOPCs overlap several epigenetic marks that also change, and most are DNA methylation. However, there is no analysis of this number in relationship to the total number of marks and thus the reader has no idea what these numbers represent. Are any of these overlaps significant or can this all be explained by random chance? Also, what does overlap mean, does the promoter need to be overlapped or a gene body, or is the analysis vicinity based.***

We invite the Reviewer to kindly consider this comment as directly addressed above. For sake of clarity, we have re-organized our manuscript and omitted any analysis conducted in cultured neonatal cells. Specific analysis of the DNA hydroxymethylation and RNA-Sequencing data in adult cells is now provided in the new Figures 5 and 6, with the latter including a Fisher p -value for statistical evaluation of each overlap.

- 5. The authors continue to compare 5mC with hydroxy methylation using immunofluorescence and state that hydroxy methylation is more affected than 5mC. While for my part the manuscript could have started here, or even after this analysis, can we really quantify two separate antibodies using arbitrary cutoffs like this?***

We respectfully note that the intention of the original figure was not to compare the immunoreactivity to antibodies specific for 5mC with that to antibodies specific for 5hmC, as this would be imprecise due to the different affinity of each antibody. Rather, we intended to provide an internal comparison of the distribution of each mark in OLIG2+ nuclei at distinct time points, comparing the immunoreactivity of each antibody in neonatal and adult samples. The reorganized data are now shown in the new Figure 1.

6. *The authors then move into their mouse experiments using several crosses and the experiments that show a role for TET1 and not TET2 in adult myelination. These experiments are much better controlled and show that global reduction of hydroxymethylation interferes with differentiation during remyelination.*

We appreciate the comment as we did use several controls in our studies, including genotype, age and isoform specificity.

7. *Then the authors use reduced representation hydroxy methylation profiling and RNA Seq to analyze their system further. There's a difference in the number of transcripts that are up and downregulated pre and post lesion in TET mutant cells (less differences) compared to controls. Again, several numbers are thrown at the reader lacking context and statistics. For instance 5583 genes are associated with hydroxy DMRs of 2kb with at least two cpgs in the gene region. How are DMRs coupled to genes, how is the gene region defined. Do longer genes that are typically overrepresented in neuronal genes hold more DMRs and is this controlled for? Is a single base shift sufficient to call a region DMR? (two for the 5583 genes ?).*

We appreciate this comment and, in response to the Reviewer's concern, we have now included a more detailed explanation of the analysis and of the results, in the new Figures 5 and 6 and in the associated Supplemental Figure 8. Besides the additional explanation in the Methods section, we have also included a paragraph in the results to better explain the rationale of the RRHP technique and address quality controls (including the number of CpGs detected per sample, the levels of DNA hydroxymethylation in each samples, the overall distribution of the differentially hydroxymethylated regions across the genome and the attribution to specific gene features). In addition, we explain that differential hydroxymethylated regions (DhMR) were defined by the presence of a minimum of 2 concordantly hydroxymethylated CpGs (in the same direction) within 2 kb in the genome. For the gene attribution, all RRHP sites that fell within 2000 bp upstream and 500 bp downstream from the TSS of a given gene, were attributed to the promoter, while the gene body linked to that gene was used to define the gene region. While in our original submission we did not control for gene size, we now provide a correlation (Supplemental Figure 8). For RNA-Sequencing, we used the same pipeline and same cut-off for all datasets, to identify differentially expressed genes (p value < 0.01 and q value < 0.05).

8. *The overlap between increased hydroxy methylated regions and upregulated transcripts is 323. Is this significant? what does this number tell us? A significant correlation would be expected between altered hydroxy methylation and altered expression. Is altered hydroxy methylation linked to the promoter regions of these genes ?*

In agreement with the Reviewer's suggestion, the overall reorganization of the manuscript focused on adult OPC and myelin repair. We have now added a new Figure 6. In panel 6D, we now show the overlap between: (i.) 1,726 genes with higher transcripts and also higher DNA-hydroxymethylation levels at promoter (1,595, from Figure 5H) and/or gene body (1,721, from Figure 5I) in aOLs compared to aOPCs and (ii.) 2,369 genes differentially upregulated in controls (but not in *Tet1* mutants) during myelin repair (from Figure 6B). In panel 6E, we show a similar analysis but for: (i.) 1,142 genes with lower transcripts and higher DNA-hydroxymethylation levels at promoter (1,004, from Figure 5H) and/or gene body (1,126, from Figure 5I) in aOLs compared to aOPCs and (ii.) 1,359 genes differentially downregulated in controls (but not in *Tet1* mutants) during myelin repair (from Figure 6C). We used the Fisher test in R (`fisher.test()`\$p.value in R, universe = 15,614), to verify that our new "442 gene-of-interest overlap" was indeed significant ($p=2.202774e^{-33}$).

9. *Slc genes are overrepresented, but these are large genes (Slc12a2 is ~200 kbp). Is this corrected for?*

In our initial analysis, we had not controlled for the gene size. However, in the new version we have included a correlation analysis in the new Supplemental Figure 8. We also would kindly note that, while some transporters are above 100kbp, the *Slc12a2* gene in mice is 68kbp and the higher level of DNA hydroxymethylation in aOLs compared to aOPCs during myelin repair does not appear to result from over-representation of genes with large gene size.

Reviewer #2 (Remarks to the Author):

Moyon et al. characterized oligodendrocyte (OL) lineage-specific conditional KO mice of TET1, the primary enzyme responsible for DNA hydroxymethylation in the CNS, and defined its role in remyelination. Combined with analyses of diverse OL and OPC-specific RNAseq datasets, the authors suggest the link between reduced TET1 levels and reduced remyelination efficiency that are commonly found in the aged CNS. The authors compared the RNAseq results from control and Tet1 KO mice, and proposed some TET1 target genes including SLC12A2, and showed that a loss of function mutant of slc12a2b, a homolog in zebrafish led to

oligodendroglial defects. With these results, Moyon et al. concluded that TET1 regulates oligodendrocyte differentiation and remyelination. The DNA hydroxymethylation mediated by the TET family may be an important epigenetic mechanism for adult oligodendroglial regulation, and this study seems to be the first to assess the contribution of this class of genes to normal myelination and adult myelin repair. However, the phenotypes of the mutant mice were somewhat weak and may require additional data to support the authors' conclusion that Tet1 critically regulates adult remyelination after demyelinating injury, and may provide the more detailed cellular mechanism. I would recommend some revisions to address some questions and ultimately improve the paper.

- 1. There are concerns about mice used for OPC and OL isolation and subsequent RNAseq, and Olig1-Cre mice because of their transgene expression (or activity) outside target cells or lineages. Therefore, additional supplemental figures should be prepared to show their target cell specificity at the examined ages. Relevant fractional quantification should be performed, and results interpretation should be cautiously made.*

In agreement with the Reviewer's request for specificity, we have now included a new Supplemental Figure 3, which addresses the loss of TET1 in oligodendroglial lineage cells, but not in neurons or astrocytes. In addition, we would like to underscore that the main focus of the manuscript is the role of TET1 in myelin repair, a finding that was validated using the conditional inducible line *Pdgfra-creERT* as shown in Figure 4. In response to the characterization of the reporter mouse lines used in the analysis, we have now included an additional Supplemental Figure 7, which describes the *Pdgfra-H2BEGFP* and the *Plp-EGFP* lines.

a. The PDGFRa-EGFP (knock-in mice with a sacrifice of one allele of Pdgfra) is known to express EGFP not only in OPCs but also in PDGFRa+ pericytes (Kang et al., 2010) or leptomeningeal cells (Marques et al. 2016; Marques et al., 2018. Therefore, the fraction of EGFP+ cells that express other than NG2+ or Olig2+ cells at the studied ages (P6, P60, and P540).

Indeed, the *PDGFRa-H2BEGFP* has been shown to be expressed in other cell types *in vivo*. By performing a Percoll gradient, prior to sorting, we were able to enriched our preparation in oligodendroglial lineage cells, while discarding blood vessel and leptomeninges. A transcriptional and immunocytochemical validation of the sorted cells is now shown Supplemental Figure 7, and identifies the sorted cells as composed of over 90% progenitor cells.

b. The Plp-EGFP Tg mice used in this study, (if these were generated by Wendy Macklin group) are known to express EGFP in NG2+ OPCs in addition to oligodendrocytes (OLs) (Mallon et al., 2002). If this line of mice were used, EGFP expression in OLs and OPCs should be carefully quantified for the target ages (P16, P60, P540) in the target brain areas. Also, the ref. (#53) does not appear to be the correct source for Plp-EGFP mice. #53 (Spassky et al., 1998) reported the generation of Plp-LacZ Tg mice.

The mice used in this study were not the ones generated by the Macklin group, but rather those generated by Spassky et al., 1998 as *Plp-LacZ* reporters and then used to generate the *Plp-EGFP* reporter in LeBras et al., 2005. Both references have been listed in the manuscript. To further address the concern of the reviewer we have now included a transcriptional and immunocytochemical analysis on the purity of the preparation. Our sorted *Plp-EGFP+* population is characterized at 92.6% MOG+ and 93.2% MBP+, while only a small percentage of the cells (7.3%) was immunoreactive for progenitor markers. These data are now included in the new Supplemental Figure 7.

c. Olig1-Cre (knock-in with a sacrifice of one allele of Olig1) was reported to target most astrocytes in addition to oligodendrocyte lineage (De Biase et al., 2011). Images of cre reporter (e.g., Ai14) expression after crossing Olig1-Cre should be shown, and if the Cre activity is broadly observed other than OL lineage, the results should be stated accordingly.

We agree with the reviewer and acknowledge that our *Olig1-cre* could be targeting other cell types. However, we show validation that TET1 expression was not significantly decreased in motoneurons and in astrocytes, and that mutants and controls were characterized by similar expression of key neuronal and astrocytic markers (new Supplemental Figure 3). In addition, as previously mentioned, we also confirmed our main results in an additional inducible OPC-specific cell line (*Pdgfra-creERT*) (Figure 4).

- 2. The authors show that oligodendroglial Tet1 (Olig1-Cre; Tet1 f/f) ablation did not impact developmental myelination (Figure 4). Even when remyelination defects were observed in the same cKO (Figure 5), the numbers (or percentage) of CC1+ cells and Olig2+ cells did not decrease (Supplemental Figure 4), indicating that the aOPC > OL differentiation is not disturbed by the Tet1 ablation. Combined together, the inefficient recovery of myelin (based on weaker MBP immunoreactivity, fluoromyelin patterns, and reduced percentage of newly myelinated axons (14 dpi) shown in Figure 5)*

suggest that Tet1 is required for myelinogenesis (or myelin maintenance), but not for oligodendrogenesis during the regenerative process. However, the authors state “Our results identify DNA hydroxy-methylation as necessary for aOPC differentiation”. Change the statements or provide supporting evidence.

Indeed, as noted by the reviewer, the main point of this paper is that DNA hydroxymethylation catalyzed by TET1 is essential for the activation of a transcriptional program responsible for the regulation of myelin repair, not oligodendrogenesis. In support of this statement, we clearly show that ablation of *Tet1* impairs new myelin formation, but does not affect the number of OLIG2 and CC1 cells. In addition, we note that genes whose expression is regulated by hydroxymethylation during myelin repair do not include myelin genes, but rather lipid metabolism and cell-cell communication. This suggests that DNA hydroxymethylation is a mechanism allowing coordinated expression of enzymes necessary for lipid metabolism and molecules regulating space at the axon-myelin interface. The title, text and figures have been revised to clarify this concept throughout the manuscript.

3. *In Figure 3, does the inefficient myelin repair observed in old mice involve impaired differentiation of OPC to OLs, or only inefficient myelination from newly formed OLs?*

Inefficient myelin repair detected in old mice compared to young, is associated with an overall sluggish response to the demyelinating injury. The data in the new Figure 1 (Figure 1I and 1K) reveal that old mice, consistent with the age-dependent decline of *Tet1*, are unable to efficiently hydroxymethylate DNA, and yet this does not impact their ability to proliferate (as shown in Supplemental Figure 1G) and reach a similar cell density as wild type by 21 days post lesion (as shown in Supplemental Figure 1F). However, old mice not only show a decreased number of myelinated axons (as in Figure 1G), but are characterized by the presence of swellings at the axon-myelin interface (as shown in Figure 7F-G), a phenotype which is also detected in young *Tet1* mutants (as shown in Figure 8E-F). Overall these data suggest that the inefficient myelin repair in old mice, characterized by aberrant axon-myelin interface, is consequent to declined levels of TET1, leading to inappropriate hydroxymethylation and expression of genes regulating this space, possibly leading to myelin instability.

4. *What is the cell stage in which TET1 exerts its effect? The authors assumed that the cell stage for Tet1 activity is OPC throughout the manuscript. But there is little or no evidence suggesting that TET1 impact on OPCs, or ‘aOPC to OL differentiation’. As pointed above, only myelination defects were seen during remyelination in the Tet1 mutant mice. If Tet1 ablation impacts only myelination without impaired new OL generation, Tet1 may play roles in OLs for myelin (membrane) growth or myelin maintenance. The authors should clearly distinguish the two possible scenarios. Can Tet1 or 5hmC be co-immunostained with NG2 and CC1, not only with Olig2, during remyelination process?*

As stated above, in the *Tet1* mutants, we did not detect significant changes in the ability of progenitors to proliferate (as shown in Supplemental Figure 6B) or form OLIG2+ /CC1+ cells (as shown in Supplemental Figure 6C). However, we detected an aberrant transcriptional response occurring during myelin repair, so that the newly formed myelin is “unstable”, and characterized by “swellings” at the axon-myelin interface (Figures 7 and 8). As such, we believe that TET1 activity is critical during the formation of new myelin membrane, when the newly formed OL have to “match” the expression of solute carriers and other genes involved in cell-cell communication, with the expression of genes involved in lipid metabolism.

5. *In Figure 6, the degree of Slc12a2 expression reduction in Tet1 cKO mice is much greater in the spinal cord tissue (Figure 6E) than in FACS-isolated OPCs (Figure 6F). Does this indicate that Tet1 ablation more impacts OLs (or other cells) than in OPCs?*

As the Reviewer accurately and acutely notices, several lines of experimental evidence argue in favor of a role for TET1 in newly formed oligodendrocytes. However, the levels of *Slc12a2* should be discussed within the framework of TET1 target genes. In other words, *Slc12a2* transcripts and SLC12A2 protein nicely correlate with the levels of *Tet1* transcripts and TET1 protein. When TET1 expression is high (e.g. in the spinal cord of young adults during myelin repair, as shown in Fig. 2C and 2F), the levels of SLC12A2 are high (Fig. 7D). When TET1 expression is low (e.g. in the TET1 mutants as in Fig. 3 and Suppl. Fig. 3 or in old mice, due to age-related decline as in Fig. 2E-F), then also the levels of SLC12A2 are low (as shown in Fig 8-D and in Supplemental Fig. 6F).

6. *In Figures 7D and 8C, after LPC, does the upregulation of Slc12a2 occur only in oligodendroglia, or in astrocytes and microglia too? What are the dashed lines (Are they expressed in gray matter)? The upregulated Slc12a2 signals during remyelination should be examined with cell stage markers (NG2 and CC1) and cell type-specific markers (astrocytes and microglia).*

While we cannot exclude the expression of SLC12A2 by other cell types, we have analyzed currently available transcriptomic and proteomic datasets. Below, we present a synopsis of the data from distinct transcriptomic and proteomic studies in cultured ad sorted cells, all supporting the enrichment of this gene in oligodendrocyte lineage cells, especially at the stage of newly formed oligodendrocytes.

Slc12a2 - Mus musculus

Zhang et al., 2014

Brain cell gene expression data

Enter gene symbol:

Data Source:

Log2 transform?

References:
Sharma K, et al. Nat Neurosci 2015; doi:10.1038/nn.4166
Darmas S, et al. PNAS 2015; doi:10.1073/pnas.1507125112
Marques S, et al. Science 2016; doi:10.1126/science.1254643
Tasic B, et al. Nat Neurosci 2016; doi:10.1038/nn.4216
Zeisel A, et al. Science 2015; doi:10.1126/science.1254643

Sharma et al., 2015

Brain cell gene expression data

Enter gene symbol:

Data Source:

Log2 transform?

References:
Sharma K, et al. Nat Neurosci 2015; doi:10.1038/nn.4166
Darmas S, et al. PNAS 2015; doi:10.1073/pnas.1507125112
Marques S, et al. Science 2016; doi:10.1126/science.1254643
Tasic B, et al. Nat Neurosci 2016; doi:10.1038/nn.4216
Zeisel A, et al. Science 2015; doi:10.1126/science.1254643

Zeisel et al., 2015

Brain cell gene expression data

Enter gene symbol:

Data Source:

Log2 transform?

References:
 Sharma K, et al. Nat Neurosci 2015; doi:10.1038/nn.4160
 Darmanis S, et al. PNAS 2015; doi:10.1073/pnas.1507125112
 Marques S, et al. Science 2016; doi:10.1126/science.1246463
 Tasic B, et al. Nat Neurosci 2016; doi:10.1038/nn.4216
 Zeisel A, et al. Science 2015; doi:10.1126/science.1258143

Tasic et al., 2016

Brain cell gene expression data

Enter gene symbol:

Data Source:

Log2 transform?

References:
 Sharma K, et al. Nat Neurosci 2015; doi:10.1038/nn.4160
 Darmanis S, et al. PNAS 2015; doi:10.1073/pnas.1507125112
 Marques S, et al. Science 2016; doi:10.1126/science.1246463
 Tasic B, et al. Nat Neurosci 2016; doi:10.1038/nn.4216
 Zeisel A, et al. Science 2015; doi:10.1126/science.1258143

Marques et al., 2016

Proteomic data:

Enter gene symbol:

Data Source:

Log2 transform?

References:
 Sharma K, et al. Nat Neurosci 2015; doi:10.1038/nn.4160
 Darmanis S, et al. PNAS 2015; doi:10.1073/pnas.1507125112
 Marques S, et al. Science 2016; doi:10.1126/science.1246463
 Tasic B, et al. Nat Neurosci 2016; doi:10.1038/nn.4216
 Zeisel A, et al. Science 2015; doi:10.1126/science.1258143

Sharma et al., 2015

As such, we believe that these data further support our interpretation on the necessity of TET1 function in newly generated oligodendrocytes to activate the expression of genes regulating myelin stability during repair.

- In Figure 8, although zebrafish *slc12a2b* is a homolog of *Slc12a2*, it is a paralog of *slc12a2a* but not the ortholog of *Slc12a2*. How about the phenotype of loss of function mutant of *slc12a2a*? Does *Tet1* also regulate hydroxymethylation of *slc12a2b* in zebrafish?**

It has been shown (Abbas & Whitfield., Development 2009) that the *Slc12a2a* zebrafish mutant presented little ears, due to the loss of endolymph, but no myelin phenotype (either in PNS and CNS). This could also be explained by compensation by *Slc12a2b* in this case (but not sufficient in the *Slc12a2b* zebrafish mutant). We have no data about the hydroxymethylation of *Slc12a2b* expression in zebrafish.

- 8. It is not clear whether the EM results of *Tet1* mutation (Figure 8E) correspond to the abnormalities (shown in Figure 8I) out of *slc12a2b* mutation in zebrafish, as one came from EM and the other from low-power magnified fluorescence. It will be helpful to show the phenotypes observed in zebrafish is similar or identical to the mouse at EM levels.**

Since the initial submission, the group of David Lyons has performed EM analysis on peripheral axons in *Slc12a2b* zebrafish mutants, which showed similar myelin abnormalities in the PNS (Marshalls-Phelps et al., 2020, Supplemental Figure S5, shown below). Because of the extensive similarities between the abnormalities observed by confocal imaging in the PNS (Marshalls-Phelps et al., 2020) and CNS (this manuscript), we believe the phenotype at the axon-myelin interface, is strikingly similar.

Figure S5.

Periaxonal space swelling, axonal enlargement, and myelin outfoldings in *slc12a2b^{ue58}* mutants. (A) TEM images of the pLLn in control (left) and *slc12a2b^{ue58}* mutant animals showing an enlarged axon (asterisk) with periaxonal swelling (middle) and myelin outfoldings (arrowheads; right). Scale bar, 1 μ m.

- Moreover, the phenotype of the mutant in zebrafish appeared during normal early development. However, *TET1* KO conditional mutant mice did not show any abnormalities in myelination. Explain this.**

This comment appears a bit harsh, as in no way we have attempted to make a direct correlation between zebrafish development and the role of hydroxymethylation in myelin repair. It is important to note that we carefully characterized the role of TET1 in mouse (not zebrafish) developmental myelination and in mouse myelin repair. The reason we included the zebrafish data is not to create a direct parallel between an enzyme and one of its gene-target, but rather to provide an example for the functional role of one of the TET1 target genes. A careful comparative analysis of the role of hydroxymethylation in developmental myelination in mouse and zebrafish we believe is outside the scope of the current study.

- 9. In Figure 8, the *mbp* promoter was used to express a fluorescence protein in zebrafish, which allowed for the label of both peripheral and central myelin. Thus, are those abnormalities shown in Figure 8G- I related to Schwann cells and OLs. In ref # 57, the co-authors reported this phenotype as peripheral myelin swelling. It would be great if the authors clearly state how this experiment results (Figure 8) are different from those of ref # 57. Are the myelin defects in zebrafish mutants caused by cell-autonomous or non-cell autonomous mechanisms?**

All the zebrafish data shown in this manuscript are observed in the CNS and we have clarified this in our new manuscript. Our reference #57 (Marphalls-Phelps et al., 2020) was used to discuss that, in the zebrafish model, a similar myelination defect was observed in both PNS and CNS.

Minor:

- 10. Grammar error (in Page 10. first paragraph) “Demyelination was induced in the spinal cord of inducible *Tet1* mutants at P60, one week after tamoxifen induction by gavage (Figure 5I). and Control mice, lacking *cre* expression, were subject to the same experimental protocol.” Remove ‘and’ between the two separate sentences.**

The entire text has been extensively edited.

- 11. What brain areas of *Tg* mice did the authors collect to isolate OPCs and OLs using FACS?**

For sorting, the entire forebrain was used.

12. In Figure 6A, in the upper diagram, P60 control mice is also denoted as *Olig1-Cre; Tet1 flox*, which is the same as mutant mice. Correct this.

For the sake of clarity, we have modified our schematic in the new figures and indicate the genotype of control mice and *Tet1* mutants.

13. In Figure 7A, provide separate each colored images, as least for the magnified view, while the merged image remains. Consider that the NFM (neuro-filament M) label to be white font in black background or use a similar alternative.

We have now added the single-color image and modified NFM legend (new Figure 7A).

14. In Figure 7D, are the number of *Olig2+* cells smaller in old mice? Then the low level of *Slc12a2* is not caused by the decreased expression, but by the decreased OPC recruitment?

We did not detect any difference in the number of OLIG2 during repair in young and old mice (new Figure 1). As such, the difference in SLC12A2 might not be only explained by a decreased number or size of OLIG2+ cells. Moreover, our new quantification of SLC12A2 in OLIG2+ cells also revealed decreased SLC12A2 expression in cells within the lesion, which would not be consistent with decreased recruitment (new Figure 7D-E).

15. In Figure 7E, there are no error bars in the white column (young age), while the bar has multiple values (dots)?

We have corrected this error. The multiple dots reflect the average for each mouse.

16. Page 12 – 13: The authors use terms somewhat inconsistently: neuroglia interface, axo-glial interface, axo-glial junction, axon-myelin junction, axon-myelin interface (page 16). ‘Axo-glial junction’ is a specialized term that describes cell adhesion flanking node of Ranvier and involves the interactions of multiple proteins. It is not clear if the provided EM data indicate disruption of axo-glial junction.

We have uniformed the use of the term “axon-myelin interface” throughout the text of the revised manuscript.

17. The authors stated: “We identified the gene ontology category “neuronal-oligodendroglial communication”, which included solute carrier transporters. We focused on the solute carrier gene family, especially on the member *Slc12a2*.” Provide GO term number for the major GOs in the text or figure legends.

We appreciate the reviewer’s request for accuracy of GO number. However, similar genes are often present within categories with different name and number, which provides confusion and complexity. In order to achieve both, clarity and precision, we have simplified the GO categories. For example the category including the Solute carrier family belongs to the category named “regulation of cell communication” in Fig 5D; the same name is also used in Fig 5G to refer to the gene category “regulation of cell processes”, and to the category “cell homeostasis” in Fig. 6B. To facilitate the reference to the correct GO category, as per Reviewer’s request we have included these numbers in the Figure legends.

18. (Page 16) “The detection of SLC12A2 at the axon-myelin interface was consistent with the concept that this is an electrically silent ionic transporter whose function is to regulate ionic balance and cell volume.” If there is a reference that showed NKCC1 regulates OL (or myelin sheath) volume, please cite it.

There are several reports on the role of NKCC1 in regulating cell volume in epithelial cells and those references were included in the initial submission (Haas and Forbush, 2000; Hamann et al., 2010). As requested by the Reviewer, we have now also included additional more recent references supporting the role of NKCC1 in regulating cell volume in the mammalian brain (Zhang et al., 2016; Zhang et al., 2020) and in the peripheral myelin sheath (Marshall-Phelps et al., 2020).

19. A recent study (Jantzie et al., 2015) suggests that inhibition of NKCC1 benefits oligodendrocyte (or WM preservation or repair after injury) in a mouse model of PVL, and so appears to be conflicting to the current report. Consider discussing this paper.

The Reviewer brings up an interesting point regarding the complex role of NKCC1 in oligodendrocyte progenitors in pathological conditions of the developing brain, such as periventricular leukomalacia. This led us to consider the inclusion of a more expansive list of papers supporting the expression of *Slc12a2* (NKCC1) in newly formed oligodendrocytes and a broad discussion on the regulation of this transporter in oligodendroglial lineage cells in physiological and pathological conditions.

20. *The title of some figures and subheadings are far moved from objective data, so somewhat subjective or not consistent with data. In Supplemental Figure 3, the title is “Constitutive ablation of Tet1 in OPC mimics the events leading to defective myelin regeneration detected in old mice.” However, the data did not show any abnormalities in the three assessed parameters (percentages of CC1+ cells, Olig2+ cells, and remyelinated axons) after Olig1-Cre-mediated Tet1 cKO. It is not clear about what events are related to defective myelination.*

We have modified the titles of Supplemental Figures in our revised manuscript to be more accurately descriptive of the content.

Reviewer #3 (Remarks to the Author):

The manuscript by Moyon et al identified a novel and necessary role of TET1-mediated DNA hydroxymethylation in adult myelin repair. The authors investigated the three TET family enzyme levels change with age, and they found DNA hydroxymethylation and increased expression of Tet1 were detected in young adult mice after demyelination, but not in the old mice. Specific ablation of Tet1 but not Tet2 led to the age-related decrease of DNA hydroxymethylation and inefficient remyelination. The authors also performed genome-wide hydroxymethylation and transcriptomic analysis to identify the Tet1 target genes including several members of the solute carrier (Slc) gene family. They found that transcript levels of Slc genes including Slc12a2 were lower in Tet1 mutants and old mice and were also associated with swelling at the neuroglial interface detected in zebrafish mutants. Overall, this work is well-executed, and the data is presented logically.

Main comments:

1. *The authors showed experimentally that compared to Tet1, Tet2 does not play a major role in adult remyelination. However, it is questionable how did the authors rule out Tet3. The reasons the authors provided are ‘Tet3 levels showing only minimal changes with age (Figure 3M)’ and ‘we were unable to detect TET3 in OPCs or OLs in the adult mouse brain, using multiple approaches. This finding is consistent with previous reports on TET3 expression limited to embryonic tissues.’ These are just not true. First, according to Figure 3M, Tet3 levels showed the largest changes with age in OPCs (smallest ANOVA p-value), even stronger than Tet1. Second, again according to Figure 3M, Tet3 is clearly detectable in adult OPCs, showing higher expression than Tet1. Figure 2F, on the other hand, showed Tet3 is barely detectable in OLIG2+, seems to be contradictory to Figure 3M. However, 2F and 3M are different techniques and measurements. I suggest cross-validating 2F and 3M with the other method. Third, on the contrary, many reports showing Tet3 is was the most expressed Tet enzyme in mouse brain (e.g. doi.org/10.1093/nar/gkq684), which agrees with the data showing in Figure 3M. Therefore, the authors need to explain convincingly why they eliminated Tet3 or to conduct the experiments for Tet3 as they did for Tet1 and Tet2.*

We agree with the Reviewer that our experimental approach has not been designed to test or entirely rule out a role for TET3 in the oligodendrocyte lineage. However, despite previous reports on tet3 expression in the forebrain, we were unable to detect TET3 in adult spinal cord samples. In this revised manuscript we have included additional in Figure 1 to show the region-specific expression of TET3 in adult OPC in brain and spinal cord. We have also toned down the interpretation of the results within the text.

2. *It is very interesting that the depletion of Tet1 only affects adult myelination but not developmental myelination. The authors mentioned very briefly DNMT might be the reason. I think more discussion should be included to show why it is the case.*

We share with the Reviewer the idea that a differential role of hydroxymethylation in developmental versus adult myelination is an important concept shown by our study. In agreement with the Reviewer's suggestion we have now extensively reorganized introduction, results and discussion to highlight differences between the epigenetic changes occurring in nOPC and regulating developmental myelination and those regulating the formation of aOL from aOPC, whose nuclei are characterized by higher levels of DNA methylation than their neonatal counterpart. In addition, we discuss the concept of DNA methylation as repressive epigenetic modification regulating the differentiation of OPC into OL and hydroxymethylation as an activating epigenetic mark coordinating the expression of solute carriers and lipid metabolism, thereby regulating the stability of the axon-myelin interface and effective myelin repair after demyelination

3. *The authors used RRHP to profile the genome-wide hydroxymethylation but did not show any results from RRHP directly. RRHP is not the most commonly used technique to sequence 5hmC. It will be good to add a supplementary figure to show*

the general analysis from RRHP, for example, the 5hmC distribution is consistent with previous 5hmC sequencing in mouse brain obtained with other methods.

We have added more details in the results, methods and in the new Figures 5 and 6 and Supplemental Figure 8. For RRHP, we briefly describe the method in the result section, discuss the quality controls and address the distribution of the differentially hydroxymethylated regions to gene features, noting the consistency with what previously reported (Song et al., 2011; Mellen et al., 2012; Lister et al., 2013) for the presence of this mark at promoter and gene body regions. Differential hydroxymethylated regions (DhMR) were identified as 2Kb regions with a minimum of 2 concordant differentially hydroxymethylated CpGs. For the gene body analysis, all RRHP sites that fell within a gene body were linked to that gene. For promoters, a flanking region of 2000 bp upstream from the gene's transcription start site to 500 bp downstream from the TSS was used as the region for pairing.

Minor comments:

- 1. There is no need to add the hyphen in 'methyl-cytosine' and 'hydroxy-methylation'.***

Noted and revised accordingly.

- 2. Page 3, "These enzymes act as methyl-cytosine dioxygenases, which convert a hydrogen atom at the C5-position of cytosine to a hydroxy-methyl group (5hmC), through oxidation of 5mC" is chemically incorrect. It should be "...convert a hydrogen atom of the methyl group at the C5-position of cytosine to a hydroxyl group..."***

Noted and revised accordingly.

- 3. Images in Figure 1 should have a higher resolution.***

This Figure has been omitted in the revised submission. All the new Figures are of high resolution (600dpi), unless compression was necessary to meet the allowable limit for submission.

- 4. Page 7, "The percentage of OLIG2+ cells expressing TET1 (Supplemental Figure 1A-D) and TET2 (Supplemental Figure 1E-H) in young and old spinal...". TET1 (Supplemental Figure 1A-D) should be TET1 (Supplemental Figure 1A, B, E, and F); TET2 (Supplemental Figure 1E-H) should be TET2 (Supplemental Figure 1C, D, G, and H).***

This has been corrected in the revised manuscript.

- 5. Figure 4J, the top and bottom panels have the same label (olig1+/+; Tet1fl/fl).***

This has been corrected in the revised manuscript.

Reviewers' Comments:

Reviewer #2:

Remarks to the Author:

Moyon et al. characterized oligodendrocyte-lineage and OPC-specific TET1 cKO mice, and showed that TET1 is predominantly responsible for OL lineage hydroxymethylation during lineage progression and remyelination. Whereas no overt myelin defects were observed during development, TET1-deleted WM exhibits inefficient remyelination as aged WM does. Through RNAseq and genome-wide hydroxymethylation analyses from enriched OL lineage cells, they suggest numerous genes are regulated by TET1, including SLC12A2, that may be involved in age-related decline in the efficiency of remyelination.

In the revision, the author made some improvements by reediting the logic flow, and the main conclusions are more effectively delivered. Some supplemental figures add helpful information on cell-type specificity of the isolated cells for RNA or epi-genome analyses. However, most of the results still are somewhat weak, and do not serve as direct evidence to support the authors' arguments concerning mechanistic links among aging, TET1 activity, SLC12A2, and remyelination efficiency.

More specifically,

1) It is not clear whether age-related decline in hydroxymethylation during remyelination is casually associated with expression changes of proposed genes in the aged spinal cord (SC). The expression changes of many genes can occur via different age-related mechanisms. At least with in vitro approaches, it should be demonstrated that TET1 levels affect expression or protein activity of target genes, including *Slc12a2*, in oligodendroglia.

2) It is not clear whether the post-demyelination upregulation of SLC12A2 contributed to rapid remyelination in mice. As the authors now know, there are earlier reports suggesting that the inhibition of SLC12A2 benefited WM recovery. Assuming that SC demyelination is different from that of those WM injuries, can the authors test whether pharmacological inhibition of SLC12A2 worsen remyelination?

3) It is not clear whether swelling phenotypes seen in *Tet1* mutant mice causes or contributes to the retarded remyelination.

Other specific concerns

♣ Statistics: In general, 'ANOVA' should be clearly stated either 'one-way' or 'two-way' ANOVA. "Student's test" should be rewritten with student's t-test. Although the authors stated that they used two-way ANOVA, it is not clear what factors and whether or not their interactions were analyzed. Please specify the independent factors, when two-way ANOVA was used (age and genotype), and what aspects are different.

♣ In Figure 1H-K: It is not clear about what statistics were used for any pairwise comparisons. Given the large variance at the 21dpl group in young (P60), it doesn't look easy to reach a statistically meaningful difference. If two-ANOVA was used as stated in the method section, what are the two factors for Figure 1H? Why was One-way ANOVA not used in each graph? Did the authors use any posthoc pairwise comparison between bars in each graph? The error bars of the graphs in Figures 1, 2, and 3 (clearly except for Figure 3D) look like SD, not SEM. Please confirm it (as commented in minor concerns).

♣ In Figure 3: (F) Density of Olig2+ cells or CC1+ cells (or Olig2+CC1+ oligodendrocytes) is more appropriate parameter than % Olig2+/DPAI+ or CC1+/DAPI+ cells. If there is a loss of Olig2+ cells, % of Olig2+/DAPI may not detect the changes sensitively because the denominator is also decreased. Western blotting of myelin proteins of the forebrain will strengthen the conclusion. (H) In relation to Figure 8, state whether there are swellings or vacuoles found in axon-myelin interface in *Tet* mutant mice (without demyelination).

♣ Supplemental Figures 2 and 3: Because TET1 immunoreactivities are so granny, the data are not so convincing. To show that Olig1-Cre-mediated *Tet1* deletion does not affect astrocyte TET1 expression, it will fair to a use astrocyte-specific nuclear marker (e.g., Sox9) rather than using GFAP.

♣ Figure 4: (J) It will be great to measure g-ratios with the available electron micrograph.

♣ Page 8 (Figure 5H): 91.7% genes upregulated in adult OLs were subjected to increase hydroxymethylation in promoter. However, to fairly interpret the relationship between DNA hydroxymethylation and gene expression during OL lineage progression, reveal what % of genes that are downregulated in aOLs compared to aOPC were also characterized as increased hydroxymethylation?

1,726 genes undergo increase expression and increased hydroxymethylation during OPC to OL lineage progression, whereas 1,142 genes undergo decrease expression with increased hydroxymethylation. Shouldn't we then interpret that DNA hydroxymethylation is not necessarily correlated with increased gene expression?

♣ Supplemental Figure 7, Cell-specificity of sorted cells: Percoll gradient step may have been helpful to remove myelin fraction and some blood vessel-associated cells. However, it is still concerning that, according to RNAseq heatmap, the OPC fraction (from Pdgfra-H2EGFP) contains high levels of PECAM1+ MCAM+ endothelial cells (comparable to levels of Olig2), and the OL fraction (from Plp-EGFP) contains ALDH1L1+ astrocytes. Moreover, it is not clear how the percoll gradient method enrich oligodendroglia (over astrocytes or neurons) from LPC-injected SC in Figure 6.

♣ Figure 7: (A) SLC12A2 immunoreactivities (green) are more prominent NFM (white)+ cell compartment. (B) SLC12A2+ immunogold particles are located in axons. Are SLC12A also expressed by neurons?

♣ Figures 7D and 8C: Immunostaining of SLC12A2 expression with NG2 or GFAP will be helpful.

♣ Figure 8B: Use mouse number as the replicates, that is one mouse (not myelinated axons) should be considered as n=1.

♣ Figure 7E and Figure 8D: Why the authors' way of comparison of SLC12A2 levels after LPC are different in these two paradigms (young vs. old and with or without Tet1). If t-test is applied to Figure 7 E (like 8D), there won't be no difference between young and old groups.

♣ Figure 8E: Provide lower magnification EM images, in addition to the cropped singled axon, so that ~10 or 30% myelinated axons that contain swelling at the interface between axon and myelin are appreciated by readers.

♣ The authors should discuss phenotypes of zebrafish Slc12a2a, the true orthologue of mammalian SLC12A2.

Minor concerns:

♣ Most bar graphs in this manuscript appear to indicate mean \pm SD (not \pm SEM) except Figure 3. Please restate it in the method section and in figure legends.

♣ In some figure legends, explanation about white arrowheads are missing. (For example, supplemental Fig 1E).

♣ In Figure 2:

(D) The FACS gating needs to be shown by clearer scatter plots.

(E) At P540, the small number of EGFP+ cells may reflect decreased Pdgfra promoter activity, and thus, it is necessary to show that the EGFP+ cells are truly OPCs with immunostaining. Provide TET1 immunoreactivity on OPCs.

♣ Consider merging Figure 2C and 2F because the P60 data are currently redundantly, as it is shown in two separate graphs. Then the new data in the combined graphs (P6, P60, and P540 for TET1 and TET2) will be in parallel to Figure 2E, although 2E indicates TET levels only in OPCs.

♣ Page 5 -6: "We used immunohistochemistry to validate the specificity of Tet1 ablation in OLIG2+ cells (Supplemental Figure 4A), and not in ISLET1+ neurons (Supplemental Figure 4C-D) or GFAP+ astrocytes (Supplemental Figure 4E -F)." The text does not match with numbers of new Supplemental Figures. Maybe the authors mean supplemental figure 5. Please carefully review.

♣ Page 7, "Collectively, these results suggest that the inability of adult OPC to reach high levels of DNA hydroxymethylation, as seen in old mice and in the Tet1 constitutive mutants, did not impair the inability of aOPCs to either proliferate or differentiate, but possibly it impaired the expression of other genes critical for myelin repair."

.. he inability > the inability

♣ PdgfraCreER/+ : The JAX stock 018280 is a BAC transgenic mice line (not knock-in), so the genetic designation of PdgfraCreER/+ should be corrected.

Reviewer #3:

Remarks to the Author:

The authors have answered my questions in the revision.

One small point, in Figure 5B, the authors showed the numbers of DhMRs in different genomic features. This is not very informative, as it is related to the size of the features and the bias coverage of RRHP. For example, gene regions, including exons and introns, are much larger than promoters and enhancers, so they by chance will have more DhMRs. Similarly, RRHP is a reduced representation approach, not a whole-genome approach. It is limited by restriction enzymes cut at specific sites and biased towards to CpG-dense region, i.e. promoters, which is overrepresented in RRHP. The authors should report relatively enrichment of DhMRs over expected number by chance taking into account the two factors above. Related, in Figure 5A, the authors should replace 'Whole genome hydroxymethylation profiling' with 'reduced representation hydroxymethylation profiling'. The authors' response to Reviewer #1's comments looks fine.

Reviewers' rebuttal for manuscript NCOMMS-20-07989A

Reviewer #2's

(Remarks from the Reviewer to the Author in bold and italics):

Moyon et al. characterized oligodendrocyte-lineage and OPC-specific TET1 cKO mice, and showed that TET1 is predominantly responsible for OL lineage hydroxymethylation during lineage progression and remyelination. Whereas no overt myelin defects were observed during development, TET1-deleted WM exhibits inefficient remyelination as aged WM does. Through RNAseq and genome-wide hydroxymethylation analyses from enriched OL lineage cells, they suggest numerous genes are regulated by TET1, including SLC12A2, that may be involved in age-related decline in the efficiency of remyelination.

In the revision, the author made some improvements by reediting the logic flow, and the main conclusions are more effectively delivered. Some supplemental figures add helpful information on cell-type specificity of the isolated cells for RNA or epi-genome analyses. However, most of the results still are somewhat weak, and do not serve as direct evidence to support the authors' arguments concerning mechanistic links among aging, TET1 activity, SLC12A2, and remyelination efficiency. More specifically,

1) It is not clear whether age-related decline in hydroxymethylation during remyelination is casually associated with expression changes of proposed genes in the aged spinal cord (SC). The expression changes of many genes can occur via different age-related mechanisms. At least with in vitro approaches, it should be demonstrated that TET1 levels affect expression or protein activity of target genes, including Slc12a2, in oligodendroglia.

2) It is not clear whether the post-demyelination upregulation of SLC12A2 contributed to rapid remyelination in mice. As the authors now know, there are earlier reports suggesting that the inhibition of SLC12A2 benefited WM recovery. Assuming that SC demyelination is different from that of those WM injuries, can the authors test whether pharmacological inhibition of SLC12A2 worsen remyelination?

3) It is not clear whether swelling phenotypes seen in Tet1 mutant mice causes or contributes to the retarded remyelination.

A response to these points and specific concerns is presented below.

1) It is not clear whether age-related decline in hydroxymethylation during remyelination is casually associated with expression changes of proposed genes in the aged spinal cord (SC). The expression changes of many genes can occur via different age-related mechanisms. At least with in vitro approaches, it should be demonstrated that TET1 levels affect expression or protein activity of target genes, including Slc12a2, in oligodendroglia.

In our submitted manuscript we had shown that the TET1-target gene *Slc12a2* transcript levels were lower in cells with *Tet1* loss-of-function, thereby addressing necessity. In this amended version we also address the issue of causality, as we now provide evidence that *Tet1* gain-of-function increases the levels of *Slc12a2*. The new data are presented in Figure 8E and in Supplemental Figure 9.

2) It is not clear whether the post-demyelination upregulation of SLC12A2 contributed to rapid remyelination in mice. As the authors now know, there are earlier reports suggesting that the inhibition of SLC12A2 benefited WM recovery. Assuming that SC demyelination is different from that of those WM injuries, can the authors test whether pharmacological inhibition of SLC12A2 worsen remyelination?

It is important to note that we have not suggested that “upregulation of SLC12A2 contributes to rapid remyelination” in mice or that its pharmacological inhibition should be considered as therapeutic, as such we believe that the points raised by the Reviewer are not directly pertinent to the current study. We note that the same Reviewer had previously wanted us to include a discussion on “a recent study (Jantzie et al., 2015) suggesting that inhibition of NKCC1 benefits oligodendrocyte (or WM preservation or repair after injury) in a mouse model of PVL.” While we respect the Reviewer's focus on the role of NKCC1/SLC12A2 as therapeutic

target in neonatal injuries, such as periventricular leukomalacia, we simply selected this gene as one of the several TET1-target genes after demyelination. The main topic of this manuscript is not SLC12A2 role in demyelinating disorders of the developing brain, but rather the role of DNA hydroxymethylation mediated by TET1 in remyelination and the effect of aging on this process. As such, we believe that the experiments suggested by the Reviewer, while interesting from a translational perspective, are outside of the scope of the current study. Within the text of the manuscript, we provide a discussion of the papers supporting the expression of *Slc12a2* (*Nkcc1*) in newly formed oligodendrocytes and a broad discussion on the regulation of this transporter in oligodendroglial lineage cells in physiological and pathological conditions. We have also included the suggested citation (Jantzie et al, 2015), on the decreased neurodegeneration detected in a model of neonatal hypoxia/ischemia after treatment with the *Slc12a2* inhibitor bumetanide.

3) It is not clear whether swelling phenotypes seen in *Tet1* mutant mice causes or contributes to the retarded remyelination.

It is important to note that, also in this case, we have not claimed that “swelling phenotypes seen in *Tet1* mutant mice causes or contributes to the retarded remyelination”. Rather, we discuss the presence of swellings at the axo-myelinic interface as a phenotype characteristically detected in older mice (with low levels of *Tet1* and of *Slc12a2*), with impaired remyelination and underscore the fact that a similar phenotype at the axo-myelinic junction is also detected in young mice lacking *Tet1* in the oligodendrocyte lineage (which also have low levels of *Tet1* and *Slc12a2*). In other words, we have avoided to make generalized claims of causality, but noticed the similarity of phenotypes between older mice with low TET1 activity and young mice with genetic deletion of *Tet1* in oligodendrocyte lineage cells, thereby providing support to the concept that TET1 levels and activity are important for myelin repair in the adult brain.

Other specific concerns:

Statistics: In general, ‘ANOVA’ should be clearly stated either ‘one-way’ or ‘two-way’ ANOVA. “Student’s test” should be rewritten with student’s t-test. Although the authors stated that they used two-way ANOVA, it is not clear what factors and whether or not their interactions were analyzed. Please specify the independent factors, when two-way ANOVA was used (age and genotype), and what aspects are different.

In Figure 1H-K: It is not clear about what statistics were used for any pairwise comparisons. Given the large variance at the 21dpi group in young (P60), it doesn’t look easy to reach a statistically meaningful difference. If two-ANOVA was used as stated in the method section, what are the two factors for Figure 1H? Why was One-way ANOVA not used in each graph? Did the authors use any posthoc pairwise comparison between bars in each graph? The error bars of the graphs in Figures 1, 2, and 3 (clearly except for Figure 3D) look like SD, not SEM. Please confirm it (as commented in minor concerns).

In response to the Reviewer’s comment, we have now clarified the factors analyzed in each figure legend and clearly specified that error bar is SEM, not SD. To summarize:

Fig.1B-D: two-way, two factors: immunoreactivity and age

Fig.1H-K: one-way, only one factor: time expressed as days after lesion

Fig.2E: one-way, only one factor: age

Fig.7C: one-way, only one factor: age

Fig.7E: two-way, two factors: age and time expressed as days after lesion

Supp Fig.1B, 1D, only one factor: time expressed as days after lesion

Supp Fig.1F: two-way, two factors: age and time expressed as days after lesion

Supp Fig.1G: one-way, only one factor: time expressed as days after lesion

Supp Fig.2B, 2D, 2F, 2H: one-way, only one factor: time expressed as days after lesion.

In Figure 3: (F) Density of *Olig2*⁺ cells or *CC1*⁺ cells (or *Olig2*⁺*CC1*⁺ oligodendrocytes) is more appropriate parameter than % *Olig2*⁺/*DAPI*⁺ or *CC1*⁺/*DAPI*⁺ cells. If there is a loss of *Olig2*⁺ cells, % of *Olig2*⁺/*DAPI* may not detect the changes sensitively because the denominator is also decreased. (H) In relation to Figure 8, state whether there are swellings or vacuoles found in axon-myelin interface in *Tet* mutant mice (without demyelination).

In response to the reviewer's request, we have edited the figure to include cell density rather than percentages in both Figure 3 (panels F and O), Supplemental Figure 3 (panels 3F and 3I) and Supplemental Figure 5 (panel 5B). We have also provided western blots of myelin proteins in the revised Figure 3 (panel 3I and quantification of three experiment in 3J). We have not provided quantification of swellings as we did not detect them in the young *Tet1* mutant mice in the absence of an injury.

Supplemental Figures 2 and 3: Because TET1 immunoreactivities are so granny, the data are not so convincing. To show that Olig1-Cre-mediated Tet1 deletion does not affect astrocyte TET1 expression, it will fair to a use astrocyte-specific nuclear marker (e.g., Sox9) rather than using GFAP.

We disagree with the Reviewer on this determination and believe that collectively, the data shown in Figures 3 and 4 and in Supplemental Figures 3 and 4 provide clear and definite experimental evidence in support of *Tet1* deletion in the oligodendrocyte lineage. We also note that we provided further validation of the lineage-specific effect of *Tet1* ablation, by reporting similar experimental results obtained in two lines of mice with constitutive and inducible ablation driven by two different oligodendrocyte-specific promoters. As such, we argue that the inclusion of Sox9 staining would not substantially strengthen the main conclusions of the current manuscript.

Figure 4: (J) It will be great to measure g-ratios with the available electron micrograph.

We have now included the g-ratio measurement in Figure 4L, and also included it in Figure 3M and in Supplemental Figure 5 (panels 5F and 5I).

Page 8 (Figure 5H): 91.7% genes upregulated in adult OLs were subjected to increase hydroxymethylation in promoter. However, to fairly interpret the relationship between DNA hydroxymethylation and gene expression during OL lineage progression, reveal what % of genes that are downregulated in aOLs compared to aOPC were also characterized as increased hydroxymethylation? 1,726 genes undergo increase expression and increased hydroxymethylation during OPC to OL lineage progression, whereas 1,142 genes undergo decrease expression with increased hydroxymethylation. Shouldn't we then interpret that DNA hydroxymethylation is not necessarily correlated with increased gene expression?

We agree with the Reviewer that DNA hydroxymethylation plays a more global role in regulation of gene expression. However, we had initially emphasized its role in transcriptional activation because the RNA sequencing data in the lesioned spinal cord identified a deficit in gene activation. Nevertheless, in agreement with the reviewer's suggestion, we have modified the text accordingly throughout the paper.

Supplemental Figure 7, Cell-specificity of sorted cells: Percoll gradient step may have been helpful to remove myelin fraction and some blood vessel-associated cells. However, it is still concerning that, according to RNAseq heatmap, the OPC fraction (from *Pdgfra-H2EGFP*) contains high levels of *PECAM1+* *MCAM+* endothelial cells (comparable to levels of *Olig2*), and the OL fraction (from *Plp-EGFP*) contains *ALDH1L1+* astrocytes. Moreover, it is not clear how the percoll gradient method enrich oligodendroglia (over astrocytes or neurons) from LPC-injected SC in Figure 6.

Pasted below are our results (on the left, below are panels C and F from revised manuscript) and the cell-specific expression pattern for *Pecam*, *Mcam* and *Aldh1l1*, from published databases (on the right, below, light blue) (Zhang et al, 2014).

Based on a cumulative assessment of these data, we argue we provided a “strong” response to the initial request. We also would like to note that the reviewer’s assessment of “**high levels of Pecam and Mecam**” is not justified, as *Mcam* is not exclusively expressed in endothelial cells, but also expressed in oligodendrocyte lineage cells in published databases, *Pecam* which is more specific for endothelial cells and is expressed at very low levels in our preparation. *Aldh111* and *Gfap* are expressed also at very low levels in our preparations. As such, we are unable to provide further evidence for the specific concerns raised by the Reviewer, as they appear to be the result of a subjective interpretation of our and published results.

Figure 7: (A) SLC12A2 immunoreactivities (green) are more prominent NFM (white)+ cell compartment. (B) SLC12A2+ immunogold particles are located in axons. Are SLC12A also expressed by neurons? Figures 7D and 8C: Immunostaining of SLC12A2 expression with NG2 or GFAP will be helpful.

We appreciate the Reviewer’s interest on SLC12A2, however, as previously mentioned, this manuscript addresses the role of TET1 in age-dependent myelin regeneration after demyelination. *Slc12a2* is one of the TET1-target genes, whose expression can be detected in several cell types although its levels are higher in newly formed oligodendrocytes. Below, we present a synopsis of the data from distinct transcriptomic and proteomic studies in cultured ad sorted cells, all supporting the enrichment of this gene in oligodendrocyte lineage cells, especially at the stage of newly formed oligodendrocytes. Based on these considerations, we believe that additional immunostainings for SLC12A2 in low expressing cells may not improve or clarify the main message of the current manuscript.

Please find expression pattern of *Slc12a2* transcript and SLC12A2 protein in distinct cell types. The reference for each study is indicated on the lower right-hand corner of each graph

Zhang et al., 2014

Brain cell gene expression data

Enter gene symbol:

Data Source:
 Sharma et al. RNA Mouse (RPKM)

Log2 transform?

References:
 Sharma K, et al. Nat Neurosci 2015; doi:10.1038/nn.4160
 Darmanis S, et al. PNAS 2015; doi:10.1073/pnas.1507125112
 Marques S, et al. Science 2016; doi:10.1126/science.1254663
 Tasic B, et al. Nat Neurosci 2016; doi:10.1038/nn.4216
 Zeisel A, et al. Science 2015; doi:10.1126/science.1254663

Sharma et al., 2015

Brain cell gene expression data

Enter gene symbol:

Data Source:
 Zeisel et al. Single Cell RNA Mouse (Norm. CPM)

Log2 transform?

References:
 Sharma K, et al. Nat Neurosci 2015; doi:10.1038/nn.4160
 Darmanis S, et al. PNAS 2015; doi:10.1073/pnas.1507125112
 Marques S, et al. Science 2016; doi:10.1126/science.1254663
 Tasic B, et al. Nat Neurosci 2016; doi:10.1038/nn.4216
 Zeisel A, et al. Science 2015; doi:10.1126/science.1254663

Zeisel et al., 2015

Brain cell gene expression data

Enter gene symbol:

Data Source:
 Tasic et al. Single Cell RNA V1 Mouse (Norm. CPM)

Log2 transform?

References:
 Sharma K, et al. Nat Neurosci 2015; doi:10.1038/nn.4160
 Darmanis S, et al. PNAS 2015; doi:10.1073/pnas.1507125112
 Marques S, et al. Science 2016; doi:10.1126/science.1254663
 Tasic B, et al. Nat Neurosci 2016; doi:10.1038/nn.4216
 Zeisel A, et al. Science 2015; doi:10.1126/science.1254663

Tasic et al., 2016

Brain cell gene expression data

Enter gene symbol:

Data Source:
 Marques et al. Single Cell Oligodendrocyte RNA Mouse (Norm. CPM)

Log2 transform?

References:
 Sharma K, et al. Nat Neurosci 2015; doi:10.1038/nn.4160
 Darmanis S, et al. PNAS 2015; doi:10.1073/pnas.1507125112
 Marques S, et al. Science 2016; doi:10.1126/science.1254663
 Tasic B, et al. Nat Neurosci 2016; doi:10.1038/nn.4216
 Zeisel A, et al. Science 2015; doi:10.1126/science.1254663

Marques et al., 2016

Proteomic data:

Enter gene symbol:

Data Source:

Log2 transform?

References:
 Sharma K, et al. Nat Neurosci 2015; doi:10.1038/nn.4160
 Damme S, et al. PNAS 2015; doi:10.1073/pnas.1507125112
 Marques S, et al. Science 2015; doi:10.1126/science.1264943
 Tasic B, et al. Nat Neurosci 2016; doi:10.1038/nn.4216
 Zeisel A, et al. Science 2015; doi:10.1126/science.1257563

Notes:

Sharma et al., 2015

Figure 8B: Use mouse number as the replicates, that is one mouse (not myelinated axons) should be considered as n=1.

We believe this comment is unwarranted as, in the figure legend, we clearly provided the total number of axons counted and the total number of mice analyzed (“Data represent ratio of SLC12A2 gold particles per myelinated axons ± SEM for n=27-64 myelinated axons, quantified on 4 control and 3 mutant mice (***)p < 0.001, Student’s test”). However, in order to avoid any confusion, we have now modified panel 8B to include the average counts for each mouse.

Figure 7E and Figure 8D: Why the authors’ way of comparison of SLC12A2 levels after LPC are different in these two paradigms (young vs. old and with or without Tet1). If t-test is applied to Figure 7 E (like 8D), there won’t be no difference between young and old groups.

We kindly invite the Reviewer to evaluate each figure within the framework of the study, rather than by simply comparing individual panels. In Figure 7E, we discuss the consequences of the overall age-dependent decrease of TET1 on the expression of its gene targets during repair after lesion (14 and 21 day-post-lesion). Figure 7E therefore addresses two parameters: age (comparing young and old) and time after lesion and uses for this reason a two-way ANOVA. Figure 8D, in contrast, simply addresses one parameter: the effect of genotype on ablation on the levels of SLC12A2. As such we use a Student’s t test to compare the level of SLC12A2 at the 14 day-post-lesion time point in *Tet1* mutant mice compared to controls. The fact that the reduction of one protein of the gene targets (SLC12A2) is greater when *Tet1* is ablated (Figure 8D) than when *Tet1* levels are reduced because of age (Figure 7E) is neither surprising or concerning, as the genetic ablation provide a greater reduction of TET1 levels.

Figure 8E: Provide lower magnification EM images, in addition to the cropped singled axon, so that ~10 or 30% myelinated axons that contain swelling at the interface between axon and myelin are appreciated by readers.

In agreement with the reviewers request we have included low magnification images of the ultramicrographs from mutants and controls and placed the single cropped axons as inserts in Figure 8 (new panel F).

The authors should discuss phenotypes of zebrafish *Slc12a2a*, the true orthologue of mammalian SLC12A2.

This point was addressed in the previous rebuttal. We have now included a paragraph in the discussion.

Minor concerns:
Most bar graphs in this manuscript appear to indicate mean ± SD (not ± SEM) except Figure 3. Please restate it in the method section and in figure legends.

We have carefully reviewed all the graphs and, for consistency, we have replaced panels to reflect error bars indicating SEM.

In some figure legends, explanation about white arrowheads are missing. (For example, supplemental Fig 1E).

We have now included explanations of the white arrowheads in all the figure legends

In Figure 2:

(D) The FACS gating needs to be shown by clearer scatter plots.

We would kindly point out that the image was highly pixelated because of the 30MB constraint for the submission of the manuscript as a pdf. Below we add the screenshot from Figure 2 at higher resolution:

(E) At P540, the small number of EGFP+ cells may reflect decreased *Pdgfra* promoter activity, and thus, it is necessary to show that the EGFP+ cells are truly OPCs with immunostaining. Provide *TET1* immunoreactivity on OPCs. Consider merging Figure 2C and 2F because the P60 data are currently redundantly, as it is shown in two separate graphs. Then the new data in the combined graphs (P6, P60, and P540 for *TET1* and *TET2*) will be in parallel to Figure 2E, although 2E indicates *TET* levels only in OPCs

In response to the Reviewer's comment on Figure 2E, we would like to note that the identity of EGFP+ as oligodendrocyte progenitors, had been addressed in the manuscript Moyon et al., Journal of Neuroscience 2015, which is referenced. In addition, we have included additional immunostaining on cell characterization in Supplemental Fig 7B. We sincerely hope that the inclusion of these will satisfy the Reviewer.

We disagree with the Reviewer regarding the merging of Figures 2C and 2F. As the text first focuses on the expression of TETs during development (Figure 2C) and then discusses the progressive impairment of the properties of OPC with time and highlights aging (Figure 2F). As such, the suggested change was not implemented.

Page 5 -6: "We used immunohistochemistry to validate the specificity of *Tet1* ablation in OLIG2+ cells (Supplemental Figure 4A), and not in ISLET1+ neurons (Supplemental Figure 4C-D) or GFAP+ astrocytes (Supplemental Figure 4E -F)." The text does not match with numbers of new Supplemental Figures. Maybe the authors mean supplemental figure 5. Please carefully review.

We apologize because the text was correct, but we had inadvertently swapped Supplemental Figure 3 and 4 when uploading the Figures. The new text reads: "We used immunohistochemistry to validate the specificity of *Tet1* ablation in OLIG2+ cells (Supplemental Figure 4A), and not in ISLET1+ neurons (Supplemental Figure 4C-D) or GFAP+ astrocytes (Supplemental Figure 4E -F)."

Page 7: "Collectively, these results suggest that the inability of adult OPC to reach high levels of DNA hydroxymethylation, as seen in old mice and in the *Tet1* constitutive mutants, did not impair the inability of aOPCs to either proliferate or differentiate, but possibly it impaired the expression of other genes critical for myelin repair."

We added the letter "t".

***PdgfraCreER/+*: The JAX stock 018280 is a BAC transgenic mice line (not knock-in), so the genetic designation of *PdgfraCreER/+* should be corrected.**

We would like to note that we did not mention BAC or knock-in neither in the results nor in the methods. Nevertheless we changed all the nomenclature to *Pdgfra-creER^(T)+/+*; *Tet1^{fl/fl}* and *Pdgfra-creER^(T)Tg^{+/+}*; *Tet1^{fl/fl}*

(Figure. 4 and Supplemental Figure 6). We kindly note that we have now followed guidelines to use +/- instead of 0/0 as it is more common and is described in: <https://www.jax.org/news-and-insights/jax-blog/2011/may/designating-genotypes-what-does-plus-really-mean20150422t150455>

Reviewer #3's point by point response:

“The authors have answered my questions in the revision. One small point, in Figure 5B, the authors showed the numbers of DhMRs in different genomic features. This is not very informative, as it is related to the size of the features and the bias coverage of RRHP. For example, gene regions, including exons and introns, are much larger than promoters and enhancers, so they by chance will have more DhMRs. Similarly, RRHP is a reduced representation approach, not a whole-genome approach. It is limited by restriction enzymes cut at specific sites and biased towards to CpG-dense region, i.e. promoters, which is overrepresented in RRHP. The authors should report relatively enrichment of DhMRs over expected number by chance taking into account the two factors above. Related, in Figure 5A, the authors should replace ‘Whole genome hydroxymethylation profiling’ with ‘reduced representation hydroxymethylation profiling’.

The authors' response to Reviewer #1's comments looks fine.”

One small point, in Figure 5B, the authors showed the numbers of DhMRs in different genomic features. This is not very informative, as it is related to the size of the features and the bias coverage of RRHP. For example, gene regions, including exons and introns, are much larger than promoters and enhancers, so they by chance will have more DhMRs. Similarly, RRHP is a reduced representation approach, not a whole-genome approach. It is limited by restriction enzymes cut at specific sites and biased towards to CpG-dense region, i.e. promoters, which is overrepresented in RRHP.

We do recognize the limitation of the RRHP approach and have included a paragraph in the text to address the issue mentioned by the Reviewer. However, we gently disagree with the determination that Figure 5B may not be informative. Indeed, the fact that gene regions are much larger than promoter and enhancers and yet, we detect increased methylation at promoter, we believe is biologically important. As such we think that the information in Figure 5B may be of interest to the readers.

Related, in Figure 5A, the authors should replace ‘Whole genome hydroxymethylation profiling’ with ‘reduced representation hydroxymethylation profiling’.

We have addressed this concern by editing the Figure, as suggested by the Reviewer.

Reviewers' Comments:

Reviewer #2:

Remarks to the Author:

In the latest revision, Moyon et al. addressed most of my concerns and clarified raised issues. With this, I think that manuscript has been improved, and now sounds more solid. This study would certainly be an interesting and important contribution to the field, and will provide important insights into age-related decline in myelin repair.

Suggestion:

- It looks like that the new results of Figure 8E (the Tet1 transduction in OPC cultures increases slc12a2 mRNAs) better fits Figure 7 (before the current Figure 7A -localization data) as it is a proof-of-principle test in vitro indicating that Slc12a2 is a target of TET1.

Reviewer #3:

Remarks to the Author:

The authors have addressed my concerns.

Reviewers' rebuttal for manuscript NCOMMS-20-07989B

We thank Reviewers 2 and 3 for helping us improve our manuscript and making it suitable for publication in Nature Communications.

Reviewer #2

(in bold and italics are the Reviewer's comments)

In the latest revision, Moyon et al. addressed most of my concerns and clarified raised issues. With this, I think that manuscript has been improved, and now sounds more solid. This study would certainly be an interesting and important contribution to the field, and will provide important insights into age-related decline in myelin repair.

We thank Reviewer 2 for their comments on our work and for helping us submitting a better manuscript.

Suggestion:

• It looks like that the new results of Figure 8E (the Tet1 transduction in OPC cultures increases slc12a2 mRNAs) better fits Figure 7 (before the current Figure 7A -localization data) as it is a proof-of-principle test in vitro indicating that Slc12a2 is a target of TET1.

As suggested, we have now moved the gain-of-function experiments, supporting *Slc12a2* as direct TET1-target gene to Figure 7. The text describing the results has been edited (p.10), and the figure legends modified accordingly.

Reviewer #3

(in bold and italics are the Reviewer's comments)

The authors have addressed my concerns.

We thank Reviewer 3 for taking the time to read and review our paper.